# Deduplication Improves Cost-Efficiency and Yields of *De Novo* Assembly and Binning of Shotgun Metagenomes in Microbiome Research

Zhiguo Zhang,[a,b,d] Lu Zhang,[b,d] Guoqing Zhang,[b,d] Ze Zhao,[b,d] Hui Wang,[b,d] Feng Ju[b,c,d,e]

aCollege of Environmental and Resources Sciences, Zhejiang University, Hangzhou, Zhejiang Province, China
bResearch Center for Industries of the Future, Key Laboratory of Coastal Environment and Resources of Zhejiang Province, School of Engineering, Westlake University, Hangzhou, Zhejiang Province, China
cCenter of Synthetic Biology and Integrated Bioengineering, School of Engineering, Westlake University, Hangzhou, Zhejiang Province, China
dInstitute of Advanced Technology, Westlake Institute for Advanced Study, Hangzhou, Zhejiang Province, China
eWestlake Laboratory of Life Sciences and Biomedicine, Hangzhou, Zhejiang Province, China

**ABSTRACT**  In the last decade, metagenomics has greatly revolutionized the study of microbial communities. However, the presence of artificial duplicate reads raised mainly from the preparation of metagenomic DNA sequencing libraries and their impacts on metagenomic assembly and binning have never been brought to attention. Here, we explicitly investigated the effects of duplicate reads on metagenomic assemblies and binning based on analyses of five groups of representative metagenomes with distinct microbiome complexities. Our results showed that deduplication considerably increased the binning yields (by 3.5% to 80%) for most of the metagenomic data sets examined thanks to the improved contig length and coverage profiling of metagenome-assembled contigs, whereas it slightly decreased the binning yields of metagenomes with low complexity (e.g., human gut metagenomes). Specifically, 411 versus 397, 331 versus 317, 104 versus 88, and 9 versus 5 metagenome-assembled genomes (MAGs) were recovered from MEGAHIT assemblies of bioreactor sludge, surface water, lake sediment, and forest soil metagenomes, respectively. Noticeably, deduplication significantly reduced the computational costs of the metagenomic assembly, including the elapsed time (9.0% to 29.9%) and the maximum memory requirement (4.3% to 37.1%). Collectively, we recommend the removal of duplicate reads in metagenomes with high complexity before assembly and binning analyses, for example, the forest soil metagenomes examined in this study.

**IMPORTANCE**  Duplicated reads in shotgun metagenomes are usually considered technical artifacts. Their presence in metagenomes would theoretically not only introduce bias into the quantitative analysis but also result in mistakes in the coverage profile, leading to adverse effects on or even failures in metagenomic assembly and binning, as the widely used metagenome assemblers and binners all need coverage information for graph partitioning and assembly binning, respectively. However, this issue was seldom noticed, and its impacts on downstream essential bioinformatic procedures (e.g., assembly and binning) remained unclear. In this study, we comprehensively evaluated for the first time the implications of duplicate reads for the *de novo* assembly and binning of real metagenomic data sets by comparing the assembly qualities, binning yields, and requirements for computational resources with and without the removal of duplicate reads. It was revealed that deduplication considerably increased the binning yields of metagenomes with high complexity and significantly reduced the computational costs, including the elapsed time and the maximum memory requirement, for most of the metagenomes studied. These results provide empirical references for more cost-efficient metagenomic analyses in microbiome research.

Address correspondence to Feng Ju, jufeng@westlake.edu.cn.

The authors declare no conflict of interest.

**KEYWORDS** duplicate reads, shotgun metagenomes, assembly, binning, microbiome, metagenomes

High-throughput DNA sequencing or next-generation sequencing (NGS) technologies have revolutionarily evolved and have been continually commercialized over the past 20 years, which nowadays enables researchers to directly obtain genomic DNA sequences of whole microbial communities (i.e., metagenomes) at an unprecedently low sequencing cost (1). The reconstruction of microbial genomes from NGS data via metagenome assembly and binning enables systematic genomic and in-depth qualitative analyses of largely uncultured microbes in environmental samples, not just the cultivable ones, offering novel insights into the microbial community functions and metabolic pathways in complex microbial systems (2, 3). Since its first application, there have been spectacular successes in recovering thousands of metagenome-assembled genomes (MAGs) for uncultured taxa from various natural and engineered ecosystems (4–6), opening the gate to the intriguing microbiome sciences underlying our earth's environment (e.g., biogeochemistry, ecorestoration, and bioremediation), bioeconomy (e.g., bioresources and bioenergy), and human systems (e.g., food and health) (7).

Duplicate reads of artificial origins, resulting mainly from the sequencing of two or more copies of the same DNA fragment amplified during PCR amplification in the library construction step, are a primary technical concern in Illumina high-throughput metagenomic sequencing (8, 9). The artificial duplicate reads do not represent any biologically meaningful information but cause the unnecessary waste of sequencing and computational resources. The removal of artificial duplicates is of critical concern in whole-genome sequencing, as duplicate reads can increase potential biases in variant-calling algorithms, in which the amplification-induced error of PCR duplicates may be misidentified as a true variant (10). The failure to remove duplicate reads can also lead to incorrect conclusions in metagenomic analyses, for example, in quantifying microbial taxa, genes, and metabolic pathways (11). The deduplication option was integrated into the standard metagenomic pipeline of MG-RAST, which conducts read-based annotation analysis (12). At the same time, it has recently been included as a quality control step in metagenomic assembly and binning tools such as the ATLAS pipeline (13) and bhattlab_workflows (https://github.com/bhattlab/), which was adopted in recent metagenomic research (14). Theoretically, the widely used metagenome assemblers and binners all need coverage information for graph partitioning and assembly binning, respectively (2, 3), where duplicate reads might result in biases in the computing coverage profile (e.g., uneven coverage), leading to adverse effects on or even failures in the metagenomic assembly and binning of genomic information for microbial species. For instance, good performance in assembling metagenomic sequencing data from an amplification-free library was observed, compared with an amplification library containing many more duplicate reads (9). However, the implications of duplicate reads in metagenomic data for downstream assembly and binning, the two core bioinformatic procedures of today's mainstream metagenomic methodology (2, 3), remain elusive.

In this study, we comprehensively evaluated for the first time the impacts of duplicate reads on the *de novo* assembly and binning of real metagenomic data sets by comparing the assembly qualities, binning yields, and requirements for computational resources with and without the removal of duplicate reads. To represent different environmental microbiomes, 45 metagenomes from five typical ecosystem habitats (i.e., human gut, bioreactor sludge, surface water, lake sediment, and forest soil) with different degrees of microbiome complexity were sampled and sequenced. We chose the widely used metaSPAdes and MEGAHIT as the assemblers and MetaBAT2 as the binner to benchmark the evaluation process. metaSPAdes is currently considered the best assembler for generating longer or better metagenome assemblies in general, whereas MEGAHIT is a fast, effective, and low-cost solution when computational resources are a limiting factor, according to previous assessment studies (15, 16).

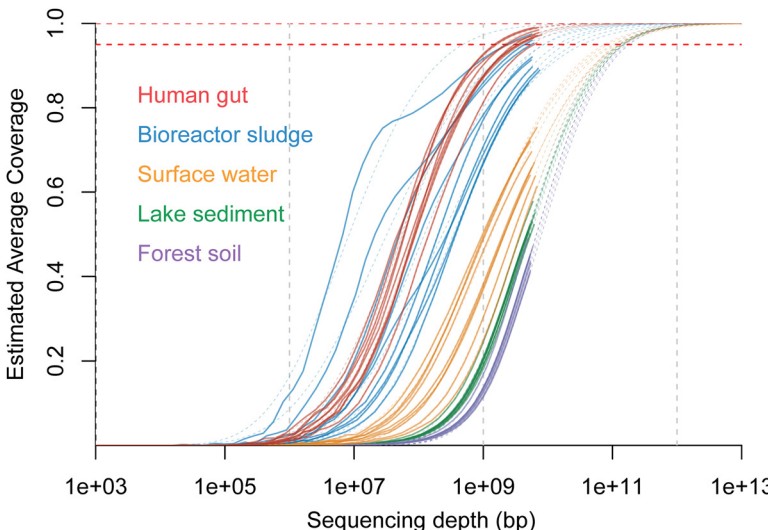

**FIG 1** Nonpareil estimates of sequence coverage (redundancy) for the 45 metagenomes studied. Metagenomes are grouped according to their environmental niche. Red, blue, orange, green, and purple lines indicate forest soil, lake sediment, surface water, bioreactor sludge, and human gut metagenomes, respectively. Solid and dashed lines indicate real curves and model predictions, respectively. The sequencing depth is displayed in base pairs, and the estimated coverage achieved is shown as a fraction of 1.

## RESULTS

**The five representative types of metagenomes used in this study.** The microbiome complexity of the metagenomes from five representative ecosystem habitats (i.e., human gut, bioreactor sludge, surface water, lake sediment, and forest soil) was evaluated using Nonpareil (17). The results confirmed that the forest soil metagenomes are the most complex among the examined metagenomic data sets, followed by lake sediment, surface water, bioreactor sludge, and human gut metagenomes (Fig. 1). On average, an affordable and regular sequencing depth of 10 Gbp for shotgun metagenomes was estimated to cover 97.0%, 91.0%, 65.4%, 49.5%, and 41.1% of the total microbiome diversity in the human gut, bioreactor sludge, surface water, lake sediment, and forest soil samples, respectively. Deduplication removed 8.8% to 25.1% of the reads from the standard data (raw data after quality filtering [see Materials and Methods]), and the duplication rate was significantly and positively correlated with the sequencing depth (see Fig. S1 in the supplemental material). Next, the standard and deduplicated data for each sample in the five groups of metagenomic data sets were assembled individually using MEGAHIT and metaSPAdes. We used four popular metrics, i.e., the number of contigs of >1 kb (1,000 bp), the number of contigs of >10 kb (10,000 bp), the longest contig length, and the $N_{50}$, to comprehensively evaluate assembly quality (Data Set S2). The results showed that assembly quality was significantly negatively correlated with microbiome complexity for both standard and deduplicated data (Fig. S2). The human gut metagenomes generally showed the highest assembly quality, followed by those of bioreactor sludge, surface water, lake sediment, and forest soil.

**Comparison of metagenomic assemblies with and without deduplication.** For MEGAHIT assemblies, deduplication considerably improved the overall length of the metagenome-assembled contigs, as the $N_{50}$ of the assemblies increased by 1.7% on average (Fig. 2a). However, deduplication reduced the total number of contigs (>1 kb) by 2.0%, 2.9%, 3.2%, 5.1%, and 7.9% on average for the human gut, bioreactor sludge, surface water, lake sediment, and forest soil metagenomes (Fig. 2a), respectively. For forest soil metagenomes, which have the highest microbiome diversity in this study, the number of contigs with a length of >10 kb significantly increased by 17.6% after deduplication (P = 0.004 by a Wilcoxon signed-rank test) (Fig. 2a).

For metaSPAdes assemblies, a significant difference in $N_{50}$ values was not observed, except for the forest soil samples, for which the $N_{50}$ was slightly lower for data with

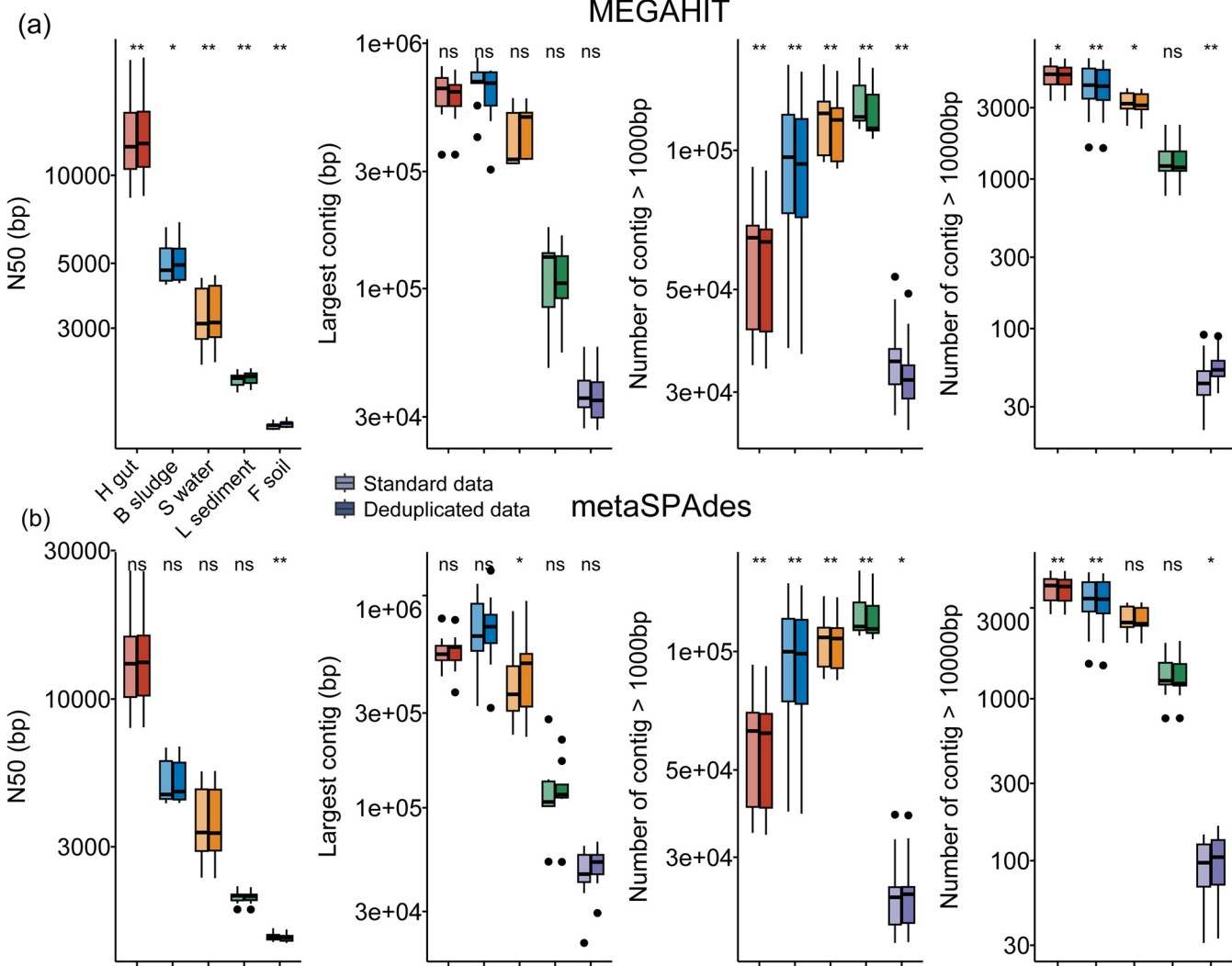

**FIG 2** Comparison of assembly results between standard and deduplicated metagenomic data. H gut, B sludge, S water, L sediment, and F soil indicate human gut, bioreactor sludge, surface water, lake sediment, and forest soil metagenomes, respectively. Significance was checked using a Wilcoxon signed-rank test. ns (nonsignificant), *, and ** indicate $P$ values of >0.05, <0.05, and <0.01, respectively.

deduplication (<1% on average) (Fig. 2b). The number of contigs with a length of >10 kb significantly ($P = 0.006$) decreased after deduplication in bioreactor sludge samples, whereas it increased in seven of nine forest soil metagenomes (Fig. 2b). In addition, the longest contig in 25 of 45 samples was longer than those without deduplication (Fig. 2f). In general, deduplication improved the length of the metagenome-assembled contigs, whereas the total number of contigs (>1 kb) was reduced by deduplication, particularly in the human gut, bioreactor sludge, surface water, and lake sediment metagenomes.

**Comparison of metagenomic binning with and without deduplication.** Deduplication contributed to the better recovery of MAGs from complex metagenomes, as revealed by binning MEGAHIT and metaSPAdes assemblies (Fig. 3) using MetaBAT2. Totals of 1,429 and 1,385 MAGs with over 50% quality (completeness − 5-fold contamination) were recovered from 45 MEGAHIT assemblies of deduplicated and standard data, respectively. In detail, there were 574 versus 578, 411 versus 397, 331 versus 317, 104 versus 88, and 9 versus 5 MAGs recovered from the human gut, bioreactor sludge, surface water, lake sediment, and forest soil metagenomes, respectively (Fig. 3a). These results showed that the number of MAGs recovered from deduplicated data was consistently higher than that recovered from standard data for all metagenomic data sets except the human gut metagenomes, which had the lowest microbiome complexity.

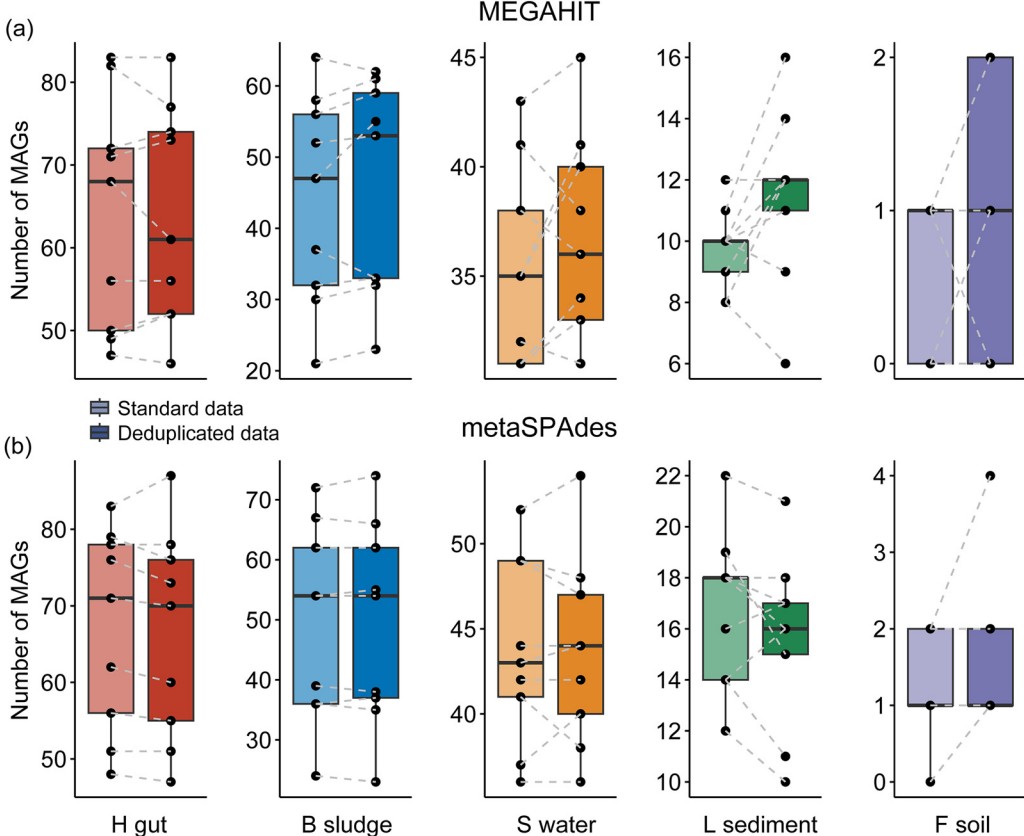

**FIG 3** Numbers of MAGs recovered from MEGAHIT and metaSPAdes assemblies of standard and deduplicated data. H gut, B sludge, S water, L sediment, and F soil indicate human gut, bioreactor sludge, surface water, lake sediment, and forest soil metagenomes, respectively.

These results also indicated that at a similar level of a 10-Gbp sequencing depth (Data Set S1), many fewer MAGs were recovered from metagenome data sets with higher microbiome complexity (Fig. 3a). Deduplication also contributed to the recovery of high-quality MAGs. For example, 11, 10, 13, and 2 more high-quality MAGs (>70% completeness and >50% quality) were recovered from the deduplicated data of the bioreactor sludge, surface water, lake sediment, and forest soil metagenomes, respectively (Table 1). Furthermore, to compare the taxonomic diversities of MAGs recovered from standard and deduplicated data, the MAGs were clustered using a 95% cutoff for whole-genome average nucleotide identity (ANI) to generate species-level representative MAGs. There were 223 versus 228, 211 versus 206, 91 versus 90, 26 versus 25, and 5 versus 2 species-level representative MAGs recovered from the human gut, bioreactor sludge, surface water, lake sediment, and forest soil metagenomes, respectively (Fig. S3). Consistent with the total number of MAGs recovered, the number of species-level MAGs recovered from the deduplicated data was also higher than that recovered from standard data for all metagenomic data sets except the human gut metagenomes, which had the lowest microbiome complexity. Moreover, most of the MAGs recovered from the standard data were also recovered from the deduplicated data. In contrast, extra species-level MAGs were recovered from the deduplicated data that were not recovered from the standard data (Fig. S4). When the taxonomic compositions of MAGs were compared, it was found that identical phylum-level taxonomic groups were recovered from the standard and deduplicated data for both human gut metagenomes and bioreactor sludge metagenomes (Fig. 4a). However, deduplication failed to recover *Myxococcota*_A MAGs from the surface water metagenomes and also failed to recover *Cyanobacteria* MAGs but instead successfully recovered *Methylomirabilota* MAGs from the lake sediment metagenomes (Fig. 4a). In contrast, the extra MAG representatives at the phylum level (i.e., *Nitrospirota* and *Actinobacteriota*)

**TABLE 1** Numbers of high-quality MAGs[a]

| Data set | No. of high-quality MAGs | | | | | |
| --- | --- | --- | --- | --- | --- | --- |
| | Deduplication with completeness of: | | | No deduplication with completeness of: | | |
| | >90% | 80–90% | 70–80% | >90% | 80–90% | 70–80% |
| MEGAHIT | | | | | | |
| Human gut | 137 | 166 | 120 | 139 | 163 | 121 |
| Bioreactor sludge | 256 | 69 | 41 | 247 | 58 | 50 |
| Surface water | 185 | 68 | 40 | 181 | 67 | 35 |
| Lake sediment | 42 | 39 | 14 | 40 | 28 | 14 |
| Forest soil | 1 | 5 | 1 | 2 | 2 | 1 |
| metaSPAdes | | | | | | |
| Human gut | 211 | 159 | 96 | 195 | 183 | 94 |
| Bioreactor sludge | 254 | 84 | 43 | 254 | 83 | 41 |
| Surface water | 173 | 103 | 59 | 171 | 99 | 64 |
| Lake sediment | 42 | 45 | 22 | 44 | 44 | 19 |
| Forest soil | 6 | 5 | 1 | 4 | 4 | 2 |

[a]The numbers of MAGs with different levels of completeness are shown. MAGs with a quality (completeness − 5× contamination) of over 50% and completeness of over 70% were counted.

were recovered from the deduplicated data but not the standard data for forest soil metagenomes (Fig. 4a), revealing the promotive effect of deduplication on genome recovery from soil metagenomes.

Totals of 1,591 and 1,602 MAGs with over 50% quality (completeness − 5× contamination) were recovered from 45 metaSPAdes assemblies of deduplicated and standard data, respectively. In detail, there were 597 versus 604, 444 versus 444, 393 versus 393, 141 versus 151, and 16 versus 9 MAGs recovered from the human gut, bioreactor sludge, surface water, lake sediment, and forest soil metagenomes, respectively (Fig. 3b). Equivalent numbers of MAGs were recovered from deduplicated and standard data for bioreactor sludge and surface water metagenomes. For forest soil metagenomes, 6 more MAGs were recovered from metaSPAdes assemblies of deduplicated data, including 2 more MAGs with >90% completeness and 1 more MAG with >80% completeness (Table 1). The number of high-quality MAGs (>70% completeness and >50% quality) was slightly higher for deduplicated data (109 MAGs) than for standard data (107 MAGs) for lake sediments (Table 1), even though 10 more MAGs were recovered from the standard data. Note that after the dereplication of MAGs (≥95% ANI), the number of species-level representative MAGs recovered from deduplicated data (597 MAGs) was almost equivalent to the number recovered from standard data (596 MAGs). There were 234 versus 241, 226 versus 223, 101 versus 96, 31 versus 33, and 5 versus 3 MAGs recovered from the human gut, bioreactor sludge, surface water, lake sediment, and forest soil data sets, respectively (Fig. S3). For the detailed composition, identical phylum-level taxonomic groups were recovered from the standard and deduplicated data for the human gut metagenomes and from the standard and deduplicated data for the surface water metagenomes (Fig. 4b). However, deduplication failed to recover *Firmicutes*_C MAGs from the bioreactor sludge metagenomes and *Cyanobacteria* MAGs from the lake sediment metagenomes (Fig. 4b). In contrast, extra phylum-level representative MAGs (i.e., *Thermoproteota* and *Krumholzibacteriota*) were recovered from the deduplicated data but not the standard data for forest soil metagenomes (Fig. 4b). Collectively, deduplication considerably promoted the binning yields of highly complex metagenomes (e.g., soil metagenomes) in both the total number of MAGs and the number of species-level MAGs, whereas it did not promote the binning yields of MAGs from metagenomes with low complexity (e.g., human gut metagenomes).

The reliance on the counts of single-copy marker genes for contamination estimation, as implemented in CheckM (18), can still lead to chimerism, with populations sometimes from entirely different clades being mixed within a single genome (19). To check the chimera levels in the MAGs constructed in this study, we annotated 10 single-copy marker genes that were validated by the mOTUs profiler (20) for their accuracy in genome

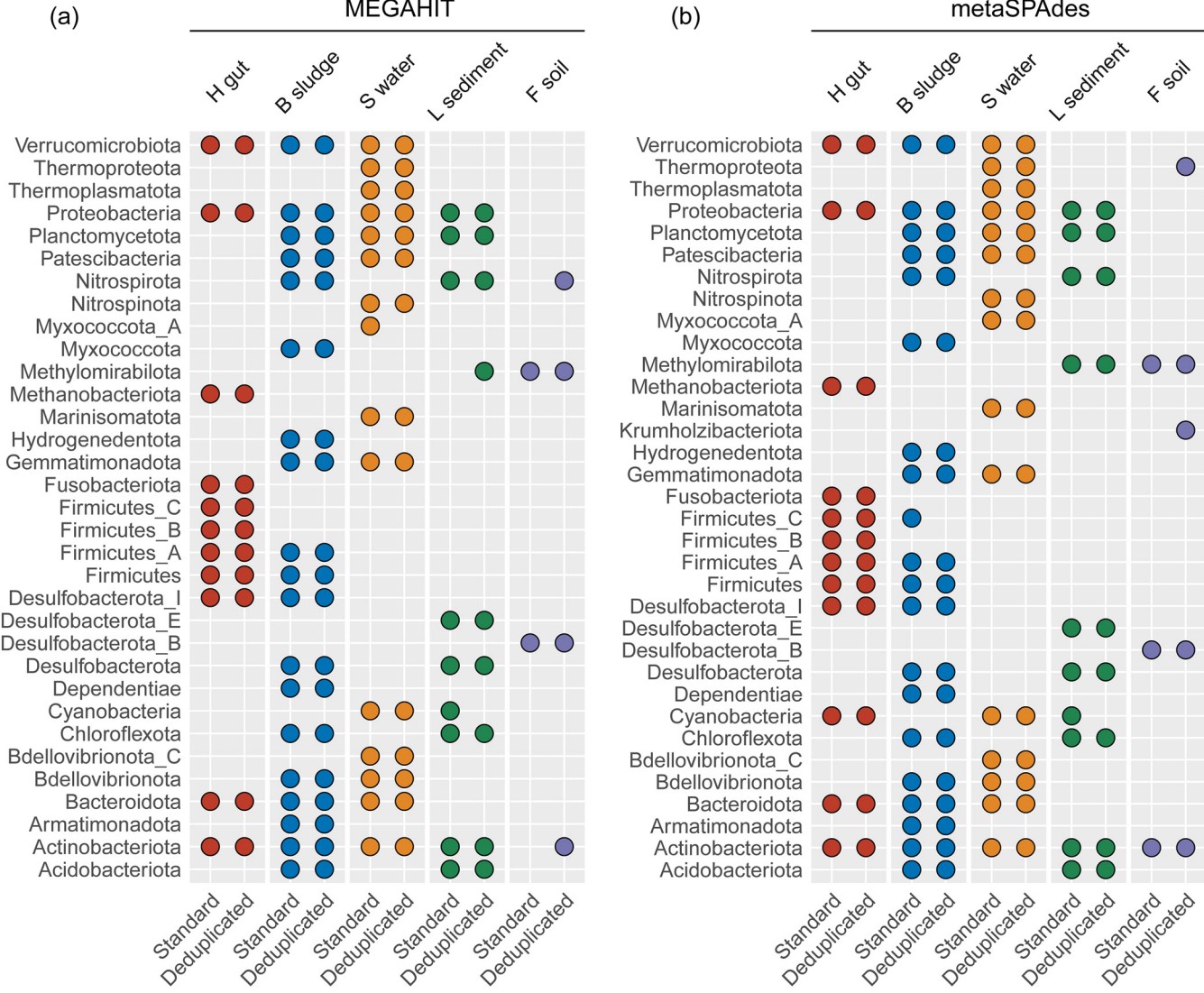

**FIG 4** Phylum-level taxonomic diversity of MAGs recovered from MEGAHIT and metaSPAdes assemblies of standard and deduplicated data. Each filled node indicates MAGs of the corresponding phylum that were successfully constructed from the related metagenomic data. H gut, B sludge, S water, L sediment, and F soil indicate human gut, bioreactor sludge, surface water, lake sediment, and forest soil metagenomes, respectively.

taxonomic classification and assessed the presence of chimeras in MAGs by evaluating the homogeneity of the taxonomic annotation for each MAG. We then compared the proportions of MAGs (disagreement rates) that contain chimeras between the metagenomic data with and without deduplication. The results showed that the disagreement rates of MAGs recovered from metaSPAdes assemblies of deduplicated data were considerably lower than those of MAGs recovered from standard data for most of the taxonomic levels (Fig. S5). The total disagreement rates of MAGs recovered from MEGAHIT assemblies and metaSPAdes assemblies for standard data and deduplicated data were 53.2% versus 49.3% and 50.5% versus 50.7%, respectively. At the genus level, the disagreement rates of MAGs recovered from metaSPAdes assemblies of deduplicated data (1.5%) were lower than those of standard data (2.5%) (Fig. S5b). These results indicate that fewer MAGs contain chimeras in deduplicated data than in standard data, particularly for metaSPAdes assemblies.

**Computational cost assessment of metagenomic assembly and binning.** Access to affordable computational resources, e.g., maximum RAM and elapsed time, are critically important for bioinformatic analysis. Therefore, we evaluated the maximum RAM and elapsed time required for the metagenomic assembly and binning of standard and deduplicated data. The results showed that deduplication significantly decreased the maximum

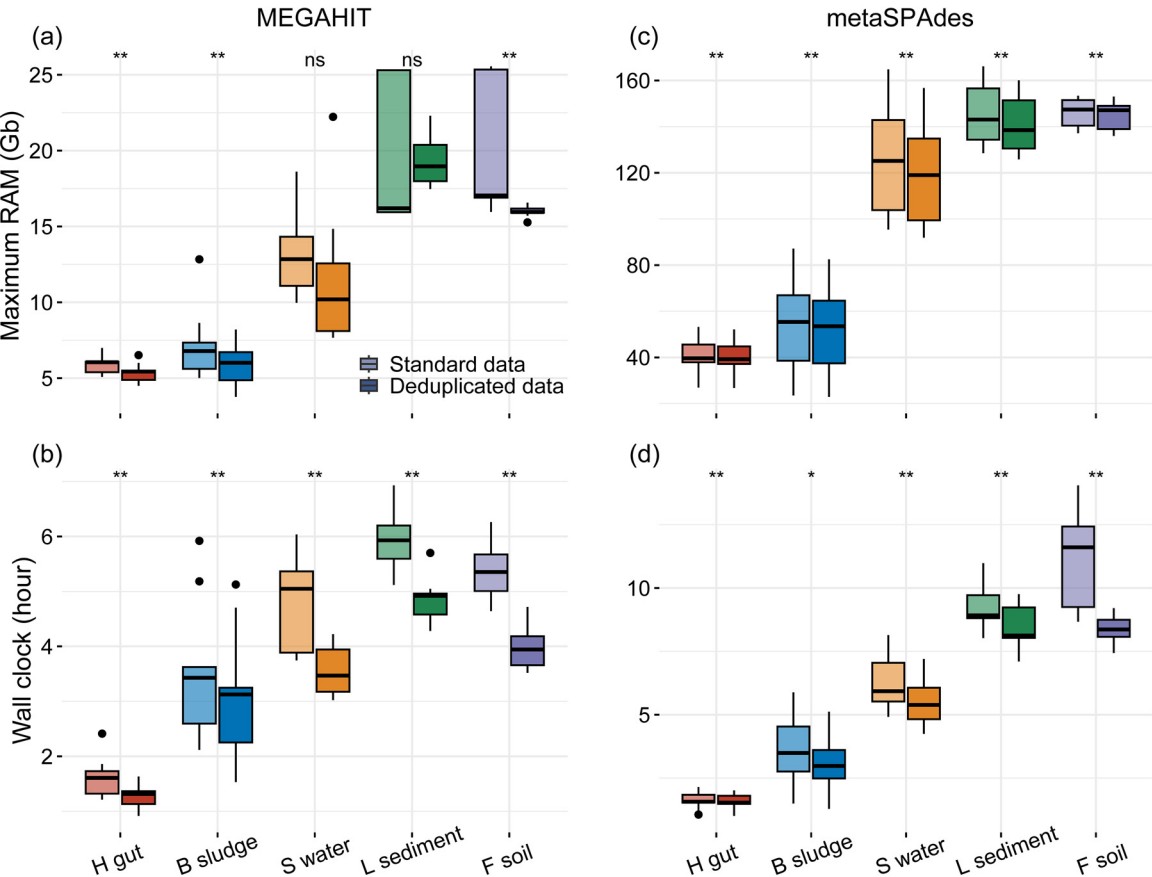

**FIG 5** Time consumption and RAM requirements for assembly of the standard and deduplicated data. Significance was checked using a Wilcoxon signed-rank test. ns, *, and ** indicate *P* values of >0.05, <0.05, and <0.01, respectively. RAM, random access memory.

memory requirement and time consumption for both MEGAHIT and metaSPAdes assemblies (Fig. 5). For example, deduplication significantly reduced the memory requirements by 9.5%, 16.7%, and 15.9% on average for the MEGAHIT assemblies of the human gut, bioreactor sludge, and forest soil metagenomes, respectively (Fig. 5a). Meanwhile, deduplication significantly reduced the elapsed time by 21.6%, 15.2%, 25.9%, 26.0%, and 18.8% for the MEGAHIT assemblies of the human gut, bioreactor sludge, surface water, lake sediment, and forest soil metagenomes, respectively (Fig. 5b). Likewise, the elapsed time was significantly reduced by 4.3%, 14.1%, 12.2%, 9.0%, and 29.9% for the metaSPAdes assemblies of bioreactor sludge, surface water, lake sediment, and forest soil metagenomes, respectively (Fig. 5d).

Deduplication also reduced the memory requirement and duration of the binning process (Fig. S6). Notably, deduplication significantly decreased the memory requirement by 13.7% and 5.9% for the binning of MEGAHIT and metaSPAdes assemblies of forest soil metagenomes, respectively (Fig. S6a and c). Simultaneously, the elapsed time was significantly reduced by 15.3%, 25.3%, 19.3%, and 28.6% for the binning of MEGAHIT assemblies of the deduplicated human gut, bioreactor sludge, surface water, and lake sediment metagenomes, respectively (Fig. S6b). Likewise, it was significantly reduced by 15.6%, 11.2%, 13.2%, and 15.0% for the binning of metaSPAdes assemblies of bioreactor sludge, surface water, lake sediment, and forest soil metagenomes, respectively (Fig. S6d). In summary, deduplication considerably reduced the elapsed time and RAM requirements in most of the studied assembly and binning cases.

## DISCUSSION

Over the last decade, improvements in next-generation metagenomic sequencing technologies and advances in computational methods have revolutionized the field of

microbiology and microbial ecology (21), which enables genome-centric studies through the recovery of metagenome-assembled genomes (MAGs) from various complex environments, including the global ocean (19), the cow rumen (22), the human gut (23, 24), aquifers (25), glaciers (26), activated sludge (27), sediment (28), and soil (29). MAGs recovered from metagenomic data allowed robust and detailed qualitative views of microbiome functional and metabolic aspects within a genomic context with high taxonomic and phylogenetic resolution (2, 3). Furthermore, the genome reconstruction of uncultivated bacteria and archaea has substantially expanded and populated the microbial tree of life (4, 30, 31) and yielded significant insights into the evolutionary relationships and metabolic properties of the uncultivated fraction of microbiomes (25). However, the presence of artificial duplicate reads (derived from library construction and paired-end sequencing) in metagenomic data and their influence on metagenomic assembly and binning have been overlooked for a long time (2).

Therefore, in this study, we systematically investigated the impacts of duplicate reads on the assembly and binning of real metagenomic data sets with various microbiome complexities, i.e., the human gut, bioreactor sludge, surface water, lake sediment, and forest soil. These results showed that deduplication improved the construction of long contigs. The size of the longest contig for most of the samples was considerably improved after deduplication, particularly for metaSPAdes assemblies (Fig. 2). The improvement of the contig length might contribute to the removal of duplicate reads in metagenomic data, which reduces the coverage bias of kmers, leading to the continuous assembly of contigs (32, 33).

Furthermore, deduplication considerably improved the binning yields in most of the studied cases. For example, the binning of MEGAHIT assemblies from deduplicated data yielded more MAGs than from standard data for bioreactor sludge, surface water, lake sediment, and forest soil metagenomes, and deduplication also increased the number of species-level representative MAGs recovered from both MEGAHIT and metaSPAdes assemblies (Fig. 3). According to previous studies, the contig coverage abundance is an important metric in the widely used binning tools (34). Therefore, duplicate reads might distort the coverage abundance profiles of contigs, which might mislead the binning tools into clustering the contigs into MAGs that they should not belong to, leading to increased contamination levels of MAGs. Our results also confirmed the decrease in the contamination levels of MAGs recovered from metaSPAdes assemblies of deduplicated data, as the results showed that deduplication reduced the disagreement rates of taxonomic annotations of MAGs (see Fig. S3 in the supplemental material). In contrast to the binning results for soil metagenomes, in which deduplication consistently promoted the binning yields of both MEGAHIT and metaSPAdes assemblies, the deduplication of human gut metagenomes slightly reduced the total binning yields of both MEGAHIT and metaSPAdes assemblies. Metagenomes with lower complexity contain more natural duplicates, which arise from genomic DNA shearing at the same position by chance in separate template molecules (35). Deduplication will remove natural duplicate reads from the predominant species of an uneven microbiota (36). It was reported that duplicate reads could account for 10 to 20% of most of the genomic data sets, 26% of which could be natural duplicates (11). For high-complexity metagenomic samples lacking dominant species, natural duplicates can make up only <1% of all duplicates (11). Therefore, due to their low microbiome complexity, more natural duplicates might exist in human gut metagenomes. Deduplication removes natural duplicates from the metagenomes, leading to decreases in the binning yields of MAGs.

Deduplication is also demonstrated to reduce the computational costs (i.e., elapsed time and RAM) of assembly and binning greatly (Fig. 5 and Fig. S6) because of the reduction of duplicate reads (by 10% to 20% in this study [Data Set S1]). Although metagenomes from different environmental habitats have similar data sizes in this study, the demands for computational resources were incredibly distinct among samples, depending on their microbiome diversity and complexity. The higher the microbiome diversity of the sample (e.g., lake sediment and forest soil) is, the longer elapsed time and more memory will be required to complete the assembly and binning. Therefore,

assembling large, complex metagenomes will incredibly challenge the computational capacity. Previously, digital normalization and partitioning of metagenomic data have been applied as preassembly filtering approaches for analyses of complex soil metagenomes, which significantly reduced the demand for computational memory and the time for metagenomic *de novo* assembly and simultaneously produced nearly identical assemblies compared with those of the unprocessed data set (37). In this study, we reveal that deduplication can decrease the demand for computational resources and improve the binning yields simultaneously, as exemplified by five distinct groups of metagenomes, i.e., the human gut, bioreactor sludge, surface water, lake sediment, and forest soil. Therefore, deduplication, digital normalization, and partitioning approaches could be coconsidered as preassembly filtering approaches in light of reducing the demand for computational resources, particularly for large, complex environmental metagenomes. Until now, several tools have been developed for the *de novo* removal of duplicate reads from metagenomes, such as Fastp (38), FastUniq (39), Fulcrum (40), and NGSReadsTreatment (41). Deduplication has recently been integrated as a quality control step in metagenomic assembly and binning tools such as the ATLAS pipeline (13) and bhattlab_workflows (https://github.com/bhattlab/). However, one of the most widely used pipelines for metagenomic assembly and binning, MetaWRAP (42), does not support the removal of duplicate reads.

Collectively, deduplication significantly promoted the assembly of long contigs and the binning yields for metagenomes with high complexity, such as the soil metagenomes. Long contigs could enable the more sensitive detection of extensive complex genomic features such as clustered regularly interspaced short palindromic repeats (CRISPRs) (43) or polyketide synthase (PKS) or nonribosomal peptide synthase (NRPS) gene clusters encoding secondary metabolites (29). The increased binning yields and binning accuracy could enable more comprehensive and precise genome-centric studies to profile microbiome functional and metabolic traits in complex microbial systems. For example, the representative MAG of *Nitrospirota*, the famous nitrite-oxidizing bacteria (44), was recovered solely from deduplicated data of forest soil metagenomes, which may promote genome-centric studies of soil nitrogen and carbon biogeochemical cycling. More importantly, the increased binning yields from deduplicated metagenomes might increase the potential for recovering previously unexplored microbial genomes. Nonetheless, there are many more natural duplicates in metagenomes with low complexity (e.g., human gut metagenomes), and deduplication will remove these natural duplicates from the metagenomes, so it is not suitable to conduct deduplication for metagenomes with low complexity, as examined in this study. Therefore, specific technologies (e.g., unique molecular identifiers) that give unique labels to each insert sequence (45) can be applied to differentiate natural duplicates and accurately remove artificial ones when desired.

## MATERIALS AND METHODS

**Sample collection and metagenomic sequencing.** Environmental samples were collected from four engineered or natural microbial ecosystems, including nine activated sludge samples from a bioreactor run in our laboratory at Westlake University (Hangzhou, China), nine surface water samples from a salt marsh in Dafeng Milu national nature preserves (Yancheng, China), nine sediment samples from Taihu Lake (Wuxi, China), and nine soil samples from Maolan Karst Forest (Libo, China). The total genomic DNA of each sample was extracted using a FastDNA spin kit for soil (MP Biomedicals, USA) according to the manufacturer's instructions and sequenced on the Illumina NovaSeq 6000 platform with a paired-end 150-bp strategy at Novogene Corporation (Beijing, China). The human gut metagenomes were randomly selected from a previous study (46) in which human gut fecal samples were collected from 942 Chinese individuals from Hong Kong and Yunnan Province, and the fecal DNA was used for metagenomic sequencing on the Illumina NovaSeq 6000 platform with a paired-end 150-bp strategy. Each sample has a data size of ~10 Gbp, and more details on the metagenomes used can be found in Data Set S1 in the supplemental material.

**Metagenomic data preprocessing, assembly, and binning.** Following metagenomic sequencing, raw reads were inspected by FastQC (v0.11.9) (47) to evaluate each metagenome's general sequence quality and PCR artifacts. The raw reads were filtered to remove low-quality reads, N-containing reads, and adapters using Fastp (v0.23.1) (38) with the following parameters: –qualified_quality_phred 5, –unqualified_percent_limit 50, and –n_base_limit 15. This yielded standard clean data (called standard data here). Next, duplicate reads in the standard data were removed using Fastp (v0.23.1) with the following parameters: –dedup –dup_calc_accuracy 3. This generated the deduplicated data (for detailed

information, see Data Set S1). To evaluate the impacts of duplicate reads on metagenomic assembly and binning, the standard data sets and deduplicated data sets were individually *de novo* assembled using both MEGAHIT (v1.2.9) (48), with the parameters –min-contig-len 1000 -m 300 -t 30, and metaSPAdes (v3.14.1) (32), with the parameters -m 400 -t 30 –only-assembler –meta. The resulting assembly for each sample was subsequently clustered into metagenome-assembled genomes (MAGs) using the MetaWRAP pipeline (42) with the following parameters: metawrap binning -o path_to_output_file binning_results -a path_to_the_assembly assembly.fasta –metabat2 -t 30 path_to_the_nine_samples/*.fastq. Briefly, the coverage profiles of each assembly were calculated by mapping the reads of every sample from the same habitat using the script jgi_summarize_bam_contig_depths (https://bitbucket.org/berkeleylab/metabat), which means that each assembly has coverage information for nine samples in this study. The coverage profiles of each assembly were then used to inform automated binning by MetaBAT2 (v2.12.1) (49) with the default parameters –minContig 1500 –maxP 95 –minS 60 –maxEdges 200 –minClsSize 200000 –minCV 1 –minCVSum 1, which is called cross-sample binning in this study. We also compared the above-described binning yields with the yields of another binning strategy (i.e., individual-sample binning) that was widely adopted in previous MAG reconstruction efforts (4, 6, 21), in which the coverage profile of each assembly was generated by mapping only reads of the corresponding sample and was used to inform binning as described above. The results revealed that cross-sample binning remarkably recovered more MAGs than individual-sample binning from all five types of metagenomes (Fig. S7). For cross-sample binning, each assembly has coverage information for all samples from the same habitat, which might contribute to the more efficient and accurate binning of the assemblies, as indicated by the binning developers (50).

**Result assessment, statistical analyses, and visualization.** The microbiome complexity of each metagenome was assessed using Nonpareil (v3.4.1) (17), a statistical program that uses read redundancy to estimate sequence coverage. The metagenomic assembly for each sample was evaluated using QUAST (v5.0.2) (51) with the following parameters: -m 1000 -t 10. The quality of the MAGs was assessed using CheckM (v1.2.0) (18), a quality assessment tool for prokaryotic genomes, with the parameters –tab_table -t 64 –pplacer_threads 24 -x fa, and MAGs with an overall quality of ≥50% (completeness − 5× contamination) were considered eligible ones. The MAGs were taxonomically annotated using the classify_wf module of gtdbtk (v1.4.0) (52) with the following parameters: –extension fa –cpus 36 –pplacer_cpus 8. Species-level representative MAGs were obtained by dereplicating the above-described MAGs using dRep (v3.0.0) (53) with the following parameters: -sa 0.95 –completeness 50 –contamination 10 –processors 36. The average nucleotide identity (ANI) was calculated using FastANI (v1.33) (54) with the following parameters: –fragLen 1000 -t 20 –matrix. The chimeras in MAGs were evaluated through taxonomic uniformity using STAG (v0.8.2 [https://github.com/zellerlab/stag]) with the following parameters: -d NCBI_10genes.stag_DB -t 24\. We annotated 10 single-copy marker genes selected from the mOTUs profiler (20). The 10 marker genes chosen have the properties of universal occurrence and a low rate of horizontal gene transfer, meaning that they have good phylogenetic properties and can be used to accurately track the evolutionary relationships among different microorganisms (20, 55). We then evaluated the homogeneity of the taxonomic annotation for each MAG, as follows: "no annotation" if fewer than two marker genes were annotated, "agreeing" if all marker genes had the same taxonomic annotation at all classification levels, and "disagreeing" if different taxonomic annotations of marker genes presented at any classification level. The disagreement rate was calculated as the number of disagreeing MAGs divided by the number of all MAGs excluding the no-annotation MAGs. The elapsed time and amount of RAM taken by the software to complete metagenomic assembly and binning were recorded using the time function in the Linux shell with the customized options \time -f %e -v -o Running.log. The figure showing estimated average coverage was generated using the Nonpareil package (17) in R (v4.0.4), and other figures were plotted using the ggplot package in R (v4.0.4). Statistical analyses were performed using the Wilcox.test function of the vegan package (56) in R (v4.0.4). Least-squares linear regression analysis was performed using the lm function of the ggplot package in R (v4.0.4).

**Data availability.** The metagenomic sequencing data used for methodology evaluation in this study were downloaded from the NCBI under BioProject accession number PRJNA588513 (human gut) and contributed from unpublished projects of laboratory mates deposited in the China National GeneBank database under accession numbers CNP0003575. The specific samples used in this study are listed in Data Set S1.

## SUPPLEMENTAL MATERIAL

Supplemental material is available online only.

**SUPPLEMENTAL FILE 1**, PDF file, 1.8 MB.

**SUPPLEMENTAL FILE 2**, XLSX file, 0.02 MB.

## ACKNOWLEDGMENTS

This work was supported by the Zhejiang Provincial Natural Science Foundation of China under grant number LR22D010001, the National Key Research and Development Program of China via project 2018YFE0110500, and the National Natural Science Foundation of China under grant number 22241603.

We thank Yisong Xu for laboratory management support. We thank the Westlake University HPC Center for computation support.

We declare no conflicts of interest.

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
