## [Reviewer comments · Microbiology Spectrum]

Microbiology Spectrum

Deduplication Improves Cost-Efficiency and Yields of De novo Assembly and Binning of Shot-Gun Metagenomes in Microbiome Research

Zhiguo Zhang, Lu Zhang, Guoqing Zhang, Ze Zhao, Hui Wang, and Feng Ju

Corresponding Author(s): Feng Ju, Westlake University

Review Timeline:

Submission Date:	November 3, 2022
Editorial Decision:	November 20, 2022
Revision Received:	December 29, 2022
Editorial Decision:	January 15, 2023
Revision Received:	January 16, 2023
Accepted:	January 18, 2023

Editor: Jianjun Wang

Reviewer(s): The reviewers have opted to remain anonymous.

Transaction Report:

DOI: <https://doi.org/10.1128/spectrum.04282-22>

November 20, 2022

Prof. Feng Ju
Westlake University
Hangzhou
China

Re: Spectrum04282-22 (Deduplication Improves Cost-Efficiency and Yields of *De novo* Assembly and Binning of Shot-Gun Metagenomes in Microbiome Research)

Dear Prof. Feng Ju:

Thank you for submitting your manuscript to Microbiology Spectrum. We have received helpful but also critical comments from two reviewers regarding your manuscript. I agree with the reviewers that this study provides a valuable comparative analysis that will be useful for the growing community of scientists using MAG binning to answer ecological questions. I also concur with the reviewers that substantial revision is needed to improve the clarity, writing, solid evidence or better discussion. For instance, a fair amount of copy editing is needed throughout the manuscript. Important and representative habitats like human guts are suggested to be considered or included. The biggest gap in the analysis itself could be a detailed comparison of the lineages of MAGs that were most considerably affected by the deduplication process. The authors are suggested to deemphasize the observed differences between metaSPAdes and MEGAHIT assembly, which have been previously reported, and refocus their efforts on explaining the features of the datasets which lead to the discrepancies. This will make the analysis more interpretable and actionable for researchers that hope to use the findings to guide their own work. The authors are also suggested to expand their evaluation of these differences in their results and discussions sections, and to expand their explanation of statistical tests performed, which appear to exist and be sufficient but are not described in the methods or supplementary file, as well as their description of the bioinformatic workflow used. Most importantly, the authors are suggested to publish codes or parameter details which would be tremendously helpful and would make the manuscript more reproducible. If you think these comments could be well replied, I am happy to have your substantially revised manuscript for reconsideration for publication.

Link Not Available

Sincerely,

jianjun wang

Journals Department
Reviewer comments:

Reviewer #1 (Comments for the Author):

Metagenomics has dramatically promoted the direct study of genetic materials of microbial communities. This manuscript systematically evaluates the impacts of artificial duplicate reads in metagenomic datasets on metagenome assembly and binning which represents the most commonly used bioinformatics procedures in today's metagenomic studies. Overall, this interesting study is rationally designed and well organized to evaluate the impacts of duplicate reads on the assembly-based metagenomic analysis, an important but still unaddressed technical issue in the field. The authors found that deduplication not only considerably increased the binning yields (by 3.5% to 80%) thanks to improved contig length and coverage profiling of contigs but also reduced the computational costs of a metagenomic assembly including elapsed time and maximum memory requirement. Based on these results, they recommended removing duplicate reads to achieve more cost-efficient processing of metagenomic data. The manuscript is well-written, and the results are logically arranged to support the major findings. The study provides valuable evaluation and new knowledge on the impact of artificially duplicated reads on the downstream metagenomic analyses and their computational cost.

Major comments:

1. The impacts of duplication were explored by analyzing metagenomic datasets of four representative ecosystem habitats, namely bioreactor sludge, surface water, lake sediment, and forest soil, which showed different levels of microbiome complexity. These samples covered typical natural or engineered environmental microbiome, while human, animal, or host environment is not evaluated. How about human gut microbiome in which the microbial diversity and complexity are generally lower? What is a potential relationship between the generation of artificial duplicates and the microbiome complexity or diversity? This can be properly discussed to expand the impact of the finding of this work.
2. The duplication level varied from 15.9% to 25.1% among the samples sequenced on the same sequencing platform by the same company. It would be insightful for the authors to dig into technical or biological underpinning of the cause of this evident level of artifacts observed. Any association between duplication level and nature of the sample or extracted DNA would be particularly useful to guide future metagenomic practice.
3. Since the results of this study have well demonstrated that dereplication is shown to contributed positively to improve qualities and reduce computational costs in the downstream metagenomic analysis, I suggested the authors to add some discussion on the potential methods or tools for removing duplication reads with emphasis on alerting the readers for tools or protocols don't have duplication in their default settings.
4. Two popular metagenome assembly tools, i.e., MEGAHIT and METASPADES, were used for the evaluation. While most results consistently suggest improved assembly and binning performance after deduplication, it seems that the selection of the two assembly tools matters for the comparative patterns before and after dereplication (e.g., Fig. 3). This should be clarified and properly discussed.
5. The idea of digital normalization and partitioning should be discussed in light of saving computational cost for metagenomic dataset.
a) Howe, Adina Chuang, Janet K. Jansson, Stephanie A. Malfatti, Susannah G. Tringe, James M. Tiedje, and C. Titus Brown. "Tackling Soil Diversity with the Assembly of Large, Complex Metagenomes." *Proceedings of the National Academy of Sciences* 111, no. 13 (April 1, 2014): 4904-9. <https://doi.org/10.1073/pnas.1402564111>.
6. The comparative results showed that binning yields of metaSPAdes assemblies are remarkably higher than the binning yields of MEGAHIT assemblies, while the disagreement rates of MAGs recovered from deduplicated data were lower than that of MAGs recovered from clean data (without deduplication) in most of taxonomic level. The interesting results seem to show that both the assembly tools and deduplication co-affect the quality of MAGs obtained, which should be but have not yet been discussed with explanations on the observed result differences between assembly tools and before and after deduplication.

Minor comments:

The fastQC should be mentioned as it is the most robust tool to check the duplication level of a metagenomic dataset.

Line 123-125: how was the coverage of microbiome diversity computed? Pls clarify

Line 131: add 'The results showed that' before 'the assembly quality'

Line 144: Use lowercase for 'Significant'

Line 145: rephrase 'lower with deduplication'

Line 166: unify the use of 'MAGs' or 'bins' which refer to the same data type

Line 171: rephrase 'Results showed' as 'The results showed' throughout

Line 198-199: how was the chimera in MAGs checked. Pls specify

Line 293: what did 'this' refer to? Pls specify

Line 567: Nonpareil should be the software used for the analysis, right? This can be briefly mentioned to ensure that all readers can easily understand what it is.

Fig. 1: X-axis 'Sequencing effort' should be 'Sequencing depth'

Fig. 2: what is the unit for the Y-axis for (c) and (d)? Pls add the label

Table 1: how was high-quality MAG defined? This should be clearly described in the legend.

Reviewer #2 (Comments for the Author):

Summary: The paper by Zhang et al. provides a systematic analysis of the effect of deduplicating sequencing reads prior to performing MAG binning. The authors show that deduplication significantly decreases computational effort in the assembly and binning process, and results in equivalent or higher numbers of MAGs across varied natural environments. This result has the potential to reduce computational effort exerted for assembly when the objective is creating MAG bins, as well as to increase recovery of novel species into MAG bins in environmental datasets.

The paper is well-composed, but would benefit from some editing for grammatical clarity, as well as some naming and organizational revision (the latter is described in more detail in the Major and Minor Comments). Overall, the most confusing aspect of the paper was the use of the two distinct assemblers across the environmental datasets. In many cases, differences between the MEGAHIT and metaSPAdes assemblers overshadowed or obscured differences attributable to the deduplication approach. The authors are encouraged to reconsider the organization of sections into MEGAHIT and metaSPAdes-specific paragraphs, and to potentially revise the focus to be on the recovery in each collection of environmental metagenomes, regardless of assembler. This would help to clarify the discrepancy that metaSPAdes results were occasionally unaffected by deduplication.

The authors are encouraged to either provide a public repository containing the code used to generate dereplication, assembly, binning, and post-analysis workflows, or to provide more detailed information on the parameters used for each tool that was applied to the data. This would ensure that the analysis is reproducible and clarify the methods. More specific notes on each tool referenced are provided in the Minor Comments below.

Major Comments:

- Beyond microbiome complexity, it is unclear how the samples varied with respect to the taxonomic diversity of MAGs, and whether specific lineages might have been disproportionately affected by your deduplication approach. I suggest that you add information about the rate of recovery of different taxonomic groups and potentially functional genes across the different environmental samples, rather than comparing the results of using two different assemblers.
- I suggest that you add a section to the discussion, in addition to identifying specific differences between the MAGs recovered with respect to captured functional and taxonomic diversity, which addresses the effect that your approach could have on existing studies. Based on the increase in taxonomic diversity of species-specific MAGs, what is the potential for extending known MAG representatives from environmental samples? This may be a more exciting note to end on than the technical methods of avoiding the presence of duplicated reads in the first place.
- No details are provided in the methods about the statistical tests that were applied to the data; this information is only included in figure legends and parenthetical remarks.

Minor Comments:

- The use of "clean" throughout to refer to quality filtered reads is confusing, since the deduplicated reads are also "cleaned." I would suggest changing the terminology to "standard" or similar to emphasize that your key advance is the deduplication, rather than the cleaning process.
- Lines 176-178: This is confusing; I think what is meant here is that most of the MAGs recovered with the standard/"clean" assembly were also recovered after deduplication of reads, but that the new MAGs generated by the deduplication process were not just duplicate species, they were new, not previously recovered species
- Lines 206-207: Consider revising language. Fewer chimeras being present implies that there were some MAGs that fully lacked disagreement, whereas here what is being assessed is a rate of disagreement on a gradient from fully distinct to "fully" chimeric (not interpretable as a distinct organism/species-level genome)
- Lines 255-256: Consider expanding this list of references; this would be more impactful if a more representative set of studies was included
- Lines 284-286: It is unclear how the duplicated reads and skewed coverage information specifically leads to contamination in this explanation. Why would it not affect only yields, beyond your observations?
- Lines 293-294: Is there a consensus between binning developers, or could this be specifically associated with the binning tool you used in the study?
- Lines 326-328: Can duplicate reads be wholly avoided? Is there a recommendation for assessment for identifying when duplicate reads may exist?
- Lines 343-344: parameters used to decide on which reads were eliminated, as well as a list of adapters, should be provided

(either in the text or in a public repository)

- Lines 350-353: for the cross-sample coverage, it is unclear what method was used to calculate coverage profiles. If this was using the companion script to MetaBAT2, this should be specified and the workflow used to generate bam coverage files should also be included
- Line 354: parameters/settings used for MetaBAT2?
- Lines 364-365: Do you expect all MAGs to be prokaryotic? Details on the filtering protocol used are missing from the sequencing methodology
- Lines 376-377: can the in-house script be published alongside the paper?
- Lines 370-372: How were these 10 marker genes selected? Can we be sure that they are balanced?
- Line 378+: Details on how figures were generated is missing; were all statistical tests and figures generated using base R utilities? If so, it would be good to explicitly specify this, as well as to list functions used in R for statistical analyses.
- Figure 2: It appears as though the significance test is being calculated across samples, yet bars are used to show the outcomes of individual assemblies for each environmental sample. This figure would be much easier to read and interpret if boxplots or other distribution-based plots could compare each set of environmental samples, rather than distinct bars that do not appear to be ordered in any particular way from left to right (unless this information is missing from the figure)
- Figure 3: A scatter plot with colors or shapes indicating each environmental sample type (along with a one-to-one line) may make it easier to read and interpret the difference in number of MAGs before and after deduplication. The two adjacent bars are similar in size and it is challenging to compare the relative difference between each sample type

Staff Comments:

Preparing Revision Guidelines

Please return the manuscript within 60 days; if you cannot complete the modification within this time period, please contact me. If you do not wish to modify the manuscript and prefer to submit it to another journal, please notify me of your decision immediately so that the manuscript may be formally withdrawn from consideration by Microbiology Spectrum.

Remarks to the Authors:

Summary: The paper by Zhang et al. provides a systematic analysis of the effect of deduplicating sequencing reads prior to performing MAG binning. The authors show that deduplication significantly decreases computational effort in the assembly and binning process, and results in equivalent or higher numbers of MAGs across varied natural environments. This result has the potential to reduce computational effort exerted for assembly when the objective is creating MAG bins, as well as to increase recovery of novel species into MAG bins in environmental datasets.

The paper is well-composed, but would benefit from some editing for grammatical clarity, as well as some naming and organizational revision (the latter is described in more detail in the Major and Minor Comments). Overall, the most confusing aspect of the paper was the use of the two distinct assemblers across the environmental datasets. In many cases, differences between the MEGAHIT and metaSPAdes assemblers overshadowed or obscured differences attributable to the deduplication approach. The authors are encouraged to reconsider the organization of sections into MEGAHIT and metaSPAdes-specific paragraphs, and to potentially revise the focus to be on the recovery in each collection of environmental metagenomes, regardless of assembler. This would help to clarify the discrepancy that metaSPAdes results were occasionally unaffected by deduplication.

The authors are encouraged to either provide a public repository containing the code used to generate dereplication, assembly, binning, and post-analysis workflows, or to provide more detailed information on the parameters used for each tool that was applied to the data. This would ensure that the analysis is reproducible and clarify the methods. More specific notes on each tool referenced are provided in the Minor Comments below.

Major Comments:

- Beyond microbiome complexity, it is unclear how the samples varied with respect to the taxonomic diversity of MAGs, and whether specific lineages might have been disproportionately affected by your deduplication approach. I suggest that you add information about the rate of recovery of different taxonomic groups and potentially functional genes across the different environmental samples, rather than comparing the results of using two different assemblers.
- I suggest that you add a section to the discussion, in addition to identifying specific differences between the MAGs recovered with respect to captured functional and taxonomic diversity, which addresses the effect that your approach could have on existing studies. Based on the increase in taxonomic diversity of species-specific MAGs, what is the potential for extending known MAG representatives from environmental samples? This may be a more exciting note to end on than the technical methods of avoiding the presence of duplicated reads in the first place.

- No details are provided in the methods about the statistical tests that were applied to the data; this information is only included in figure legends and parenthetical remarks.

Minor Comments:

- The use of “clean” throughout to refer to quality filtered reads is confusing, since the deduplicated reads are also “cleaned.” I would suggest changing the terminology to “standard” or similar to emphasize that your key advance is the deduplication, rather than the cleaning process.
- Lines 176-178: This is confusing; I think what is meant here is that most of the MAGs recovered with the standard/“clean” assembly were also recovered after deduplication of reads, but that the new MAGs generated by the deduplication process were not just duplicate species, they were new, not previously recovered species
- Lines 206-207: Consider revising language. Fewer chimeras being present implies that there were some MAGs that fully lacked disagreement, whereas here what is being assessed is a rate of disagreement on a gradient from fully distinct to “fully” chimeric (not interpretable as a distinct organism/species-level genome)
- Lines 255-256: Consider expanding this list of references; this would be more impactful if a more representative set of studies was included
- Lines 284-286: It is unclear how the duplicated reads and skewed coverage information specifically leads to contamination in this explanation. Why would it not affect only yields, beyond your observations?
- Lines 293-294: Is there a consensus between binning developers, or could this be specifically associated with the binning tool you used in the study?
- Lines 326-328: Can duplicate reads be wholly avoided? Is there a recommendation for assessment for identifying when duplicate reads may exist?
- Lines 343-344: parameters used to decide on which reads were eliminated, as well as a list of adapters, should be provided (either in the text or in a public repository)
- Lines 350-353: for the cross-sample coverage, it is unclear what method was used to calculate coverage profiles. If this was using the companion script to MetaBAT2, this should be specified and the workflow used to generate bam coverage files should also be included
- Line 354: parameters/settings used for MetaBAT2?
- Lines 364-365: Do you expect all MAGs to be prokaryotic? Details on the filtering protocol used are missing from the sequencing methodology
- Lines 376-377: can the in-house script be published alongside the paper?
- Lines 370-372: How were these 10 marker genes selected? Can we be sure that they are balanced?
- Line 378+: Details on how figures were generated is missing; were all statistical tests and figures generated using base R utilities? If so, it would be good to explicitly specify this, as well as to list functions used in R for statistical analyses.
- Figure 2: It appears as though the significance test is being calculated across samples, yet bars are used to show the outcomes of individual assemblies for each environmental sample. This figure would be much easier to read and

interpret if boxplots or other distribution-based plots could compare each set of environmental samples, rather than distinct bars that do not appear to be ordered in any particular way from left to right (unless this information is missing from the figure)

- Figure 3: A scatter plot with colors or shapes indicating each environmental sample type (along with a one-to-one line) may make it easier to read and interpret the difference in number of MAGs before and after deduplication. The two adjacent bars are similar in size and it is challenging to compare the relative difference between each sample type

Remarks to the Editor: I believe that the manuscript by Zhang et al. is a valuable comparative analysis that will be useful for the growing community of scientists using MAG binning to answer ecological questions. The manuscript needs a fair amount of copy editing, so I refrained from making comments about grammar or diction in my review. In my opinion, the biggest gap in the analysis itself is a detailed comparison of the lineages of MAGs that were most considerably affected by the deduplication process. The authors rarely point out taxonomic distinctions between recovered MAGs, in particular because their environmental samples of varying complexity had such dramatically different rates of MAG binning. I suggest that the authors deemphasize the observed differences between metaSPAdes and MEGAHIT assembly, which have been previously reported, and refocus their efforts on explaining the features of the datasets which lead to the discrepancies. This will make the analysis more interpretable and actionable for researchers that hope to use the findings to guide their own work. I suggest that the authors expand their evaluation of these differences in their results and discussions sections. The authors also need to expand their explanation of statistical tests performed, which appear to exist and be sufficient but are not described in the methods or supplementary file, as well as their description of the bioinformatic workflow used. For the latter in particular, publishing code or parameter details would be tremendously helpful and would make the manuscript more reproducible, as the major advance presented is their computational workflow. After these changes have been made, I recommend that the authors resubmit the manuscript.

Responses to the Comments of the Editor and Reviewers

Dear Editor and Reviewers,

Thanks for your precious comments on our paper entitled “Deduplication Improves Cost-Efficiency and Yields of *De novo* Assembly and Binning of Shot-Gun Metagenomes in Microbiome Research” (Manuscript ID: Spectrum04282-22). We have modified the manuscript according to your insightful and valuable comments. Please see the revision highlighted in yellow in the revised manuscript and below the point-by-point responses to the comments.

Editor’s comments:

1. We have received helpful but also critical comments from two reviewers regarding your manuscript. I agree with the reviewers that this study provides a valuable comparative analysis that will be useful for the growing community of scientists using MAG binning to answer ecological questions. I also concur with the reviewers that substantial revision is needed to improve the clarity, writing, solid evidence or better discussion.

Response: Thank you for the decision on a major revision of our work. We have revised the manuscript according to your and the reviewers’ comments and suggestions.

2. For instance, a fair amount of copy editing is needed throughout the manuscript. Important and representative habitats like human guts are suggested to be considered or included.

Response: We have edited the language throughout the manuscript. Following the suggestion, we also investigated the effects of deduplication on the assembly and binning of human gut metagenomes and included the results in the revised version of the manuscript.

3. The biggest gap in the analysis itself could be a detailed comparison of the lineages of MAGs that were most considerably affected by the deduplication process. The authors are suggested to deemphasize the observed differences between metaSPAdes and MEGAHIT assembly, which have been previously reported, and refocus their efforts on explaining the features of the datasets which lead to the discrepancies. This will make the analysis more interpretable and actionable for researchers that hope to use the findings to guide their own work.

Response: We have deemphasized the comparison between metaSPAdes and MEGAHIT assembly. We have reorganized the sections into MEGAHIT- and metaSPAdes-specific paragraphs, as suggested, to make the description more understandable, as shown in Line 140-148 and in Line 149-158.

4. The authors are also suggested to expand their evaluation of these differences in their results and discussions sections, and to expand their explanation of statistical tests performed, which appear to exist and be sufficient but are not described in the methods or supplementary file, as well as their description of the bioinformatic workflow used.

Response: We have expanded the discussion about the differences, as shown in Line 309-310, Line 311-320, and Line 321-327. The methods of statistical tests were also clarified in the Materials and methods, as shown in Line 459-461. The bioinformatic methods we used in this study were also explicitly described in the Materials and methods of the revised manuscript, as shown in Line 395-453.

5. Most importantly, the authors are suggested to publish codes or parameter details which would be tremendously helpful and would make the manuscript more reproducible. If you think these comments could be well replied, I am happy to have your substantially revised manuscript for reconsideration for publication.

Response: The parameters for the bioinformatic tools used in this study were explicitly listed in the Materials and methods of the revised manuscript, as shown in Line 395-449. Thank you for your further consideration of our manuscript.

6. When submitting the revised version of your paper, please provide (1) point-by-point responses to the issues raised by the reviewers as file type "Response to Reviewers," not in your cover letter, and (2) a PDF file that indicates the changes from the original submission (by highlighting or underlining the changes) as file type "Marked Up Manuscript - For Review Only".

Response: Thanks for your kind reminding. We have prepared the responses and the submitted files, as you suggested.

Reviewer #1:

Metagenomics has dramatically promoted the direct study of genetic materials of microbial communities. This manuscript systematically evaluates the impacts of artificial duplicate reads in metagenomic datasets on metagenome assembly and binning which represents the most commonly used bioinformatics procedures in today's metagenomic studies. Overall, this interesting study is rationally designed and well organized to evaluate the impacts of duplicate reads on the assembly-based metagenomic analysis, an important but still unaddressed technical issue in the field. The authors found that deduplication not only considerably increased the binning yields (by 3.5% to 80%) thanks to improved contig length and coverage profiling of contigs but also reduced the computational costs of a metagenomic assembly including elapsed time and maximum memory requirement. Based on these results, they recommended removing duplicate reads to achieve more cost-efficient processing of metagenomic data. The manuscript is well-written, and the results are logically arranged to support the major findings. The study provides valuable evaluation and new knowledge on the impact of artificially duplicated reads on the downstream metagenomic analyses and their computational cost.

Response: Thank you very much for the positive comments and recognition of our work. We have made point-by-point responses to your valuable comments as follows.

Major comments:

1. The impacts of duplication were explored by analyzing metagenomic datasets of four representative ecosystem habitats, namely bioreactor sludge, surface water, lake sediment, and forest soil, which showed different levels of microbiome complexity. These samples covered typical natural or engineered environmental microbiome, while human, animal, or host environment is not evaluated. How about human gut microbiome in which the microbial diversity and complexity are generally lower? What is a potential relationship between the generation of artificial duplicates and the microbiome complexity or diversity? This can be properly discussed to expand the impact of the finding of this work.

Response: Thank you for bringing up this important point. We followed your helpful suggestion and evaluated the impacts of deduplication on the metagenomic assembly and binning of human gut metagenomes. On average, the microbial complexity was lowest when compared with the other four sample types, as shown in Fig. 1. The complexity value is 0.03 for human gut samples. In contrast, it is 0.09, 0.35, 0.50, and 0.59 for bioreactor sludge, surface water, lake sediment, and forest soil, respectively. Surprisingly, the deduplication of human gut metagenomes slightly decreased the total binning yields for the binning of both MEGAHIT and metaSPAdes assemblies. We also added a detailed description of this result in Line 164-170, Line 201-204 and Line 211-217 of the revised manuscript.

Line 164-170: For details, there were 574 versus 578, 411 versus 397, 331 versus 317, 104 versus 88 and 9 versus 5 MAGs recovered from the human gut, bioreactor sludge, surface water, lake sediment, and forest soil metagenomes, respectively (Fig. 3a). The results showed that the number of MAGs recovered from deduplicated data was consistently higher than that of standard data for all metagenomic datasets, except for the human gut metagenomes which have the lowest microbiome complexity.

Line 201-204: For details, there were 597 versus 604, 444 versus 444, 393 versus 393, 141 versus 151 and 16 versus 9 MAGs recovered from the human gut, bioreactor sludge, surface water, lake sediment, and forest soil metagenomes, respectively (Fig.

3b).

Line 211-217: Note that, after the dereplication of MAGs ($\geq 95\%$ ANI), the number of species-level representative MAGs recovered from deduplicated data (597 MAGs) was almost equivalent to those recovered from standard data (596 MAGs). There were 234 versus 241, 226 versus 223, 101 versus 96, 31 versus 33 and 5 versus 3 MAGs recovered from the human gut, bioreactor sludge, surface water, lake sediment and forest soil datasets, respectively (Fig. S3).

No previous studies have reported the relationship between microbiome complexity or diversity and the generation of artificial duplicates. We observed no significant relationship between our datasets' microbiome complexity and the level of duplicate reads. However, metagenomes with lower complexity might contain more natural duplicates which arise from genomic DNA shearing at the same position by chance in separate template molecules. It was reported that the duplicate reads would account for 10-20% of most of the genomic datasets, of which 26% would be natural duplicates¹. For high-complexity metagenomic samples lacking dominant species, natural duplicates only make up <1% of all duplicates¹. There might be more natural duplicates in the human gut metagenomes due to their low microbiome complexity, and the deduplication removed the natural duplicates from the metagenomes, leading to decreased binning yields of MAGs. We also added a detailed discussion in Line 314-327 of the revised manuscript.

Line 314-327: In contrast with the binning results of soil metagenomes, in which the deduplication consistently promoted the binning yields of both MEGAHIT and metaSPAdes assemblies, the deduplication of human gut metagenomes slightly reduced the total binning yields of both MEGAHIT and metaSPAdes assemblies. Metagenomes with lower complexity contain more natural duplicates which arise from genomic DNA shearing at the same position by chance in separate template molecules (35). Deduplication will remove the natural duplicate reads from the predominant species of an uneven microbiota (36). It was reported that the duplicate reads could account for 10-20% of most of the genomic datasets, of which 26% could be natural duplicates (11). For high-complexity metagenomic samples lacking

dominant species, natural duplicates can only make up <1% of all duplicates (11). Therefore, more natural duplicates might exist in the human gut metagenomes due to their low microbiome complexity. The deduplication removed the natural duplicates from the metagenomes, leading to the decrease in binning yields of MAGs.

2. The duplication level varied from 15.9% to 25.1% among the samples sequenced on the same sequencing platform by the same company. It would be insightful for the authors to dig into technical or biological underpinning of the cause of this evident level of artifacts observed. Any association between duplication level and nature of the sample or extracted DNA would be particularly useful to guide future metagenomic practice.

Response: Thank you for bringing up the critical point. The artificial duplicate reads may consist of PCR duplicates from library preparation and optical duplicates, clustering duplicates and sister duplicates from paired-end sequencing. During the library preparation, the original genomic DNA fragments are usually amplified via PCR to create enough DNA materials and ligate adapters and barcodes to DNA fragments. PCR duplicates occur when two copies of the same original DNA fragment get onto different beads or primer lawns in a flowcell. Higher rates of PCR duplicates arise when people have too little starting DNA material, so greater amplification of the library is needed^{2,3}. Therefore, we should provide enough input DNA for library construction to have enough unique DNA fragments. In addition, PCR preferentially enriches smaller and more GC-poor molecules^{3,4}, which would also increase the duplicate rate. The duplication level is also correlated to the sequencing depth, which means that the duplication level will increase with the increasing sequencing depth (The trouble with PCR duplicates | The Molecular Ecologist). We also observed a significant positive relationship between duplication rate and sequencing depth, as shown in the following Figure. We also clarified this point in Line 129-130 of the revised manuscript.

3. Since the results of this study have well demonstrated that dereplication is shown to contributed positively to improve qualities and reduce computational costs in the downstream metagenomic analysis, I suggested the authors to add some discussion on the potential methods or tools for removing duplication reads with emphasis on alerting the readers for tools or protocols don't have duplication in their default settings.

Response: Thank you for the insightful suggestion. We have followed your suggestion and discussed the potential methods or tools for removing duplicate reads in the revised manuscript.

Line 349-355: Up to now, several tools have been developed for the de novo removal of duplicate reads from metagenomes, such as Fastp (42), FastUniq (43), Fulcrum (44), and NGSReadsTreatment (45). Deduplication has recently been included as a quality control step in the metagenomic assembly and binning tools, such as ATLAS pipeline (14) and bhattlab_workflows (<https://github.com/bhattlab/>). However, one of the most widely-used pipelines for metagenomic assembly and binning, MetaWRAP (46), does not support the removal of duplicate reads.

4. Two popular metagenome assembly tools, i.e., MEGAHIT and METASPADES, were used for the evaluation. While most results consistently suggest improved assembly and binning performance after deduplication, it seems that the selection of the two assembly tools matters for the comparative patterns before and after dereplication (e.g., Fig. 3). This should be clarified and properly discussed.

Response: Thank you for bringing up the important point. We did not intend to compare the binning yields of MEGAHIT and metaSPAdes assemblies. As both assemblers are widely used in the current metagenomic study, we would like to show the results to researchers and hopefully provide valuable information regarding the selection of assemblers. Deduplication promoted the binning yields of MEGAHIT assemblies in four sample types, while human gut and lake sediment samples are the exception in the binning of metaSPAdes assemblies. We also made the clarification in Line 186-211 of the manuscript.

5. The idea of digital normalization and partitioning should be discussed in light of saving computational cost for metagenomic dataset. a) Howe, Adina Chuang, Janet K. Jansson, Stephanie A. Malfatti, Susannah G. Tringe, James M. Tiedje, and C. Titus Brown. "Tackling Soil Diversity with the Assembly of Large, Complex Metagenomes." *Proceedings of the National Academy of Sciences* 111, no. 13 (April 1, 2014):4904-9. <https://doi.org/10.1073/pnas.1402564111>.

Response: Thank you for bringing up this important point. We have discussed these essential approaches in Line 337-349 of the revised manuscript.

Line 337-349: Therefore, assembling a big-size complex metagenome will significantly challenge the computational capacity. Previously, digital normalization and partitioning of metagenomic data have been applied as preassembly filtering approaches in the analysis of complex soil metagenomes, which could significantly reduce the demand of computational memory and time for metagenomic *de novo* assembly and simultaneously produce nearly identical assemblies as compared with the unprocessed dataset (32). In this study, we revealed that deduplication could decrease the demand for computational resources and improve the binning yields

simultaneously as exemplified by five distinct groups of metagenomes, i.e., human gut, bioreactor sludge, surface water, lake sediment and forest soil. Therefore, deduplication, digital normalization and partitioning approaches could be co-considered as preassembly filtering approaches in light of reducing the demand of computational resources, particularly for big-size complex environmental metagenomes.

6. The comparative results showed that binning yields of metaSPAdes assemblies are remarkably higher than the binning yields of MEGAHIT assemblies, while the disagreement rates of MAGs recovered from deduplicated data were lower than that of MAGs recovered from clean data (without deduplication) in most of taxonomic level. The interesting results seem to show that both the assembly tools and deduplication co-affect the quality of MAGs obtained, which should be but have not yet been discussed with explanations on the observed result differences between assembly tools and before and after deduplication.

Response: Thank you for bringing up this point. We observed that the binning yields of metaSPAdes assemblies are higher than that of MEGAHIT assemblies, and more high-quality MAGs were recovered from metaSPAdes than from MEGAHIT in all five types of metagenomes (Table 1). It was reported that the assembly quality (e.g., N50, the longest contig, the number of contigs over 10000bp) of metaSPAdes was better than that of MEGAHIT in the previous studies, which might contribute to better binning results. However, we did not observe a significant difference between assemblies of deduplicate data and no deduplicated data, except for soil metagenomes. Therefore, the reasons for the higher yields of binning metaSPAdes assemblies are still unclear.

In this study, we mainly focused on comparing the amount and quality of MAGs recovered from data with and without deduplication. We found that the disagreement rates of MAGs recovered from deduplicated data were lower than that of MAGs recovered from clean data (without deduplication) at most of the taxonomic levels. It was speculated that the presence of duplicate reads might affect the coverage

abundance of contigs, leading to the increase of contamination level of MAGs. We also provided explanations in Line 309-310 of the revised manuscript.

Minor comments:

1. The fastQC should be mentioned as it is the most robust tool to check the duplication level of a metagenomic dataset.

Response: Thank you for your suggestion. We have clarified the usage of fastQC in the Materials and methods of the revised manuscript.

Line 394-395: Following metagenomic sequencing, raw reads were inspected by FastQC (v0.11.9) (36) to evaluate the general sequence quality and PCR artifacts of each metagenome.

2. Line 123-125: how was the coverage of microbiome diversity computed? Pls clarify

Response: Thank you for bringing up this point. The coverage of microbiome diversity was estimated by Nonpareil (v3.4.1), a statistical program that uses read redundancy to estimate sequence coverage. We mentioned this method in the Materials and methods of the manuscript (Line 428-430). We also clarified it again in the place where it confused you to avoid the same confusion from future readers, as shown in Line 119-121 of the revised manuscript.

3. Line 131: add 'The results showed that' before 'the assembly quality'

Response: Thank you and revised.

4. Line 144: Use lowercase for 'Significant'

Response: Thanks for your careful check. This issue has been corrected and checked throughout the manuscript.

5. Line 145: rephrase 'lower with deduplication'

Response: Thank you for the suggestion. We have rephrased the sentence as “in which

N50 was slightly lower for data with deduplication”

6. Line 166: unify the use of 'MAGs' or 'bins' which refer to the same data type

Response: Thank you for bringing up this point. We have unified it as “MAGs”.

7. Line 171: rephrase 'Results showed' as 'The results showed' throughout

Response: Thank you. We have revised this issue throughout the manuscript.

8. Line 198-199: how was the chimera in MAGs checked. Pls specify

Response: Thank you for bringing up this point. The chimera in MAGs was evaluated through taxonomic uniformity using STAG (v0.8.2, <https://github.com/zellerlab/stag>). We annotated 10 single-copy marker genes recommended by STAG and evaluated the homogeneity of their taxonomic annotation for each MAG, as follows: “No annotation” if less than two marker genes were annotated; “Agreeing” if all marker genes had the same taxonomic annotation at all classification levels; and “Disagreeing” if different taxonomic annotation of marker genes presented at any classification level. The disagreement rate was calculated by the number of “Disagreeing” MAGs divided by the number of all MAGs excluding “No annotation”. This method was also adopted by the recent Nature paper ⁵ to check the chimera in MAGs. We have explicitly described this issue in the Materials and methods (Line 442-454) of the revised manuscript.

9. Line 293: what did 'this' refer to? Pls specify

Response: we are sorry for confusing you. “this” refers to cross-sample binning, in which each assembly has the coverage information of all samples from the same habitat rather than the single sample corresponding to the assembly. We have revised it to avoid confusion in the revised manuscript.

Line 423-426: In the across-sample binning, each assembly has the coverage information of all samples from the same habitat, which might contribute to more efficient and accurate binning of assemblies, as also indicated by binning developers (32).

10. Line 567: Nonpareil should be the software used for the analysis, right? This can be briefly mentioned to ensure that all readers can easily understand what it is.

Response: Thanks for your considerate suggestion. We have mentioned this software in the Materials and methods (Line 428-430) and the Results section (Line 119-121).

11. Fig. 1: X-axis 'Sequencing effort' should be 'Sequencing depth'

Response: Thanks for your suggestion. We have changed it to “Sequencing depth”.

12. Fig. 2: what is the unit for the Y-axis for (c) and (d)? Pls add the label

Response: Thank you for the reminding. The unit for the Y-axis of (c) and (d) is the same as the Y-axis of (a) and (b). we have added the label.

13. Table 1: how was high-quality MAG defined? This should be clearly described in the legend.

Response: we are sorry for our negligence. High-quality MAGs refer to MAGs with a quality (completeness – 5 × contamination) over 50% and with completeness over 70%. We have added a clearly description in the table legend.

Reviewer #2:

1. Summary: The paper by Zhang et al. provides a systematic analysis of the effect of deduplicating sequencing reads prior to performing MAG binning. The authors show that deduplication significantly decreases computational effort in the assembly and binning process, and results in equivalent or higher numbers of MAGs across varied natural environments. This result has the potential to reduce computational effort exerted for assembly when the objective is creating MAG bins, as well as to increase recovery of novel species into MAG bins in environmental datasets.

Response: Thank you for the positive comments on our work.

2. The paper is well-composed, but would benefit from some editing for grammatical clarity, as well as some naming and organizational revision (the latter is described in more detail in the Major and Minor Comments). Overall, the most confusing aspect of

the paper was the use of the two distinct assemblers across the environmental datasets. In many cases, differences between the MEGAHIT and metaSPAdes assemblers overshadowed or obscured differences attributable to the deduplication approach. The authors are encouraged to reconsider the organization of sections into MEGAHIT and metaSPAdes-specific paragraphs, and to potentially revise the focus to be on the recovery in each collection of environmental metagenomes, regardless of assembler. This would help to clarify the discrepancy that metaSPAdes results were occasionally unaffected by deduplication.

Response: Thanks for your insightful suggestions. The reasons we chose two distinct assemblers are that MEGAHIT and metaSPAdes were both widely used in the existing studies. We want to illustrate the effects of deduplication on both MEGAHIT and metaSPAdes assemblies and their binning results, so as to provide critical information to researchers regarding to the selection of assembler. We have reorganized the sections into MEGAHIT- and metaSPAdes-specific paragraphs, as you suggested, to make the description more understandable, as shown in Line 140-148 and Line 149-158 of the revised manuscript. Following your suggestion, we have focused the analyses on the recovery of taxonomic groups in each collection of environmental metagenomes, as shown in the following responses to your Major comments. We have also provided detailed descriptions in Line 187-198 and Line 217-224 of the revised manuscript.

3. The authors are encouraged to either provide a public repository containing the code used to generate dereplication, assembly, binning, and post-analysis workflows, or to provide more detailed information on the parameters used for each tool that was applied to the data. This would ensure that the analysis is reproducible and clarify the methods. More specific notes on each tool referenced are provided in the Minor Comments below.

Response: Thanks for your thoughtful suggestions. We have provided the detailed parameters used by each tool in the Materials and methods of the revised manuscript. For example, Line 403-410: the standard datasets and deduplicated datasets were

individually *de novo* assembled using both MEGAHIT (v1.2.9) (48) with the followed parameters: “--min-contig-len 1000 -m 300 -t 30” and metaSPAdes (v3.14.1) (33) with the following parameters “-m 400 -t 30 --only-assembler --meta”. The resulting assembly of each sample was subsequently clustered into metagenome-assembled genomes (MAGs) using the MetaWRAP pipeline (43) with the followed parameters: “metawrap binning -o path_to_output_file binning_results -a path_to_the_assembly assembly.fasta --metabat2 -t 30 path_to_the_nine_samples/*.fastq”.

Line 430-444: Metagenomic assembly of each sample was evaluated using QUAST (v5.0.2) (50) with the following parameters: “-m 1000 -t 10”. The quality of metagenome-assembled genomes (MAGs) was assessed using checkM (v1.2.0) (19) with the following parameters: “--tab_table -t 64 --pplacer_threads 24 -x fa”, and the MAGs with an overall quality $\geq 50\%$ (completeness–5×contamination) were considered as eligible ones. Species-level representative MAGs were obtained by dereplicating the above MAGs using dRep (v3.0.0) (51) with the following parameters: “-sa 0.95 --completeness 50 --contamination 10 --processors 36”. The average nucleotide identity (ANI) was calculated using FastANI (v1.33) (52) with the following parameters: “--fragLen 1000 -t 20 --matrix”. The MAG chimera in MAGs was evaluated through taxonomic uniformity using STAG (v0.8.2, <https://github.com/zellerlab/stag>) with the following parameters: “-d NCBI_10genes.stag_DB -t 24 \”.

Major comments:

- Beyond microbiome complexity, it is unclear how the samples varied with respect to the taxonomic diversity of MAGs, and whether specific lineages might have been disproportionately affected by your deduplication approach. I suggest that you add information about the rate of recovery of different taxonomic groups and potentially functional genes across the different environmental samples, rather than comparing the results of using two different assemblers.

Response: Thanks for your insightful suggestions. We have deemphasized the

comparison of two assemblers (i.e., MEGAHIT and metaSPAdes). Following your advice, we have investigated the recovery of different taxonomic groups across the different environmental samples with and without deduplication. The results showed that identical phylum-level taxonomic groups were recovered from the standard data and deduplicated data of human gut metagenomes. In contrast, the extra MAG representatives at the phylum level were recovered from deduplicated data of forest soil metagenomes compared with standard data of forest soil metagenomes, indicating the positive effects of deduplication for metagenomes with high complexity.

We also made an explicit description in Line 189-195 and Line 217-224 of the revised manuscript.

Line 189-200: When taxonomic composition of MAGs was compared, it was found that identical phylum-level taxonomic groups were recovered from the standard and deduplicated data of both human gut metagenomes and bioreactor sludge metagenomes (Fig. 4a). However, the deduplication failed to recover Myxococcota_A MAGs from the surface water metagenomes, and also failed to recover Cyanobacteria MAGs but instead successfully recovered the Methylophilota MAGs from the lake sediment metagenomes (Fig. 4a). In contrast, the extra MAG representatives at the phylum level (i.e., Nitrospirota and Actinobacteriota) were recovered from deduplicated data but not standard data of forest soil metagenomes (Fig. 4a), revealing a promotive effect of deduplication on the genome recovery from soil metagenomes.

Line 217-224: For the detailed composition, the identical phylum-level taxonomic groups were recovered from the standard and deduplicated data of human gut metagenomes and from the standard data and deduplicated data of surface water metagenomes (Fig. 4b). However, the deduplication failed to recover Firmicutes_C MAGs from the bioreactor sludge metagenomes and Cyanobacteria MAGs from the lake sediment metagenomes (Fig. 4b). In contrast, the extra phylum representative MAGs (i.e., Thermoproteota and Krumholzibacteriota) were recovered from deduplicated data but not standard data of forest soil metagenomes (Fig. 4b).

- I suggest that you add a section to the discussion, in addition to identifying specific

differences between the MAGs recovered with respect to captured functional and taxonomic diversity, which addresses the effect that your approach could have on existing studies. Based on the increase in taxonomic diversity of species-specific MAGs, what is the potential for extending known MAG representatives from environmental samples? This may be a more exciting note to end on than the technical methods of avoiding the presence of duplicated reads in the first place.

Response: Thanks for your insightful suggestions. We have added a section to address the effect of deduplication on existing studies and simultaneously reduced the discussion on technical methods of avoiding the presence of duplicated reads, as shown in Line 356-375 of the revised manuscript.

Following your suggestion, we have investigated the recovery of different taxonomic groups across the different environmental samples with and without deduplication (Line 189-195 and Line 217-224) and discussed the effects on current studies (Line 356-375). The results showed that deduplication considerably extended the recovery of the known MAG representatives from MEGAHIT and metaSPAdes assemblies of complex environmental samples, such as forest soils. For example, the representative MAG of Nitrospirota, the famous nitrite-oxidizing bacteria, was solely recovered from deduplicated data of forest soil metagenomes, which may promote the genome-centric study of soil nitrogen and carbon biogeochemical cycling.

Line 356-375: Collectively, deduplication significantly promoted the assembly of long contigs and binning yields for metagenomes with high complexity, such as the soil metagenomes. The long contigs could enable more sensitive detection of extensive complex genomic features such as clustered regularly interspaced short palindromic repeats (CRISPR) (43), polyketide synthase (PKS) or non-ribosomal peptide synthase (NRPS) gene clusters encoding for secondary metabolites (29). The increased binning yields and binning accuracy could enable more comprehensive and precise genome-centric studies to profile the microbiome's functional and metabolic traits in complex microbial systems. For example, the representative MAG of Nitrospirota, the famous nitrite-oxidizing bacteria (44), was solely recovered from deduplicated data of forest soil metagenomes, which may promote the genome-centric study of soil nitrogen and

carbon biogeochemical cycling. More importantly, the increased binning yields from deduplicated metagenomes might increase the potential for recovering previously unexplored microbial genomes. Nonetheless, there are much more natural duplicates in the metagenomes with low complexity (e.g., human gut metagenomes), and deduplication will remove these natural duplicates from the metagenomes, so it is not suitable to conduct deduplication for the metagenomes with low complexity as examined in this study. Therefore, specific technologies (e.g., Unique Molecular Identifiers), which give a unique label to each insert sequence (45), can be applied to differentiate natural duplicates and accurately remove artificial ones when desired.

- No details are provided in the methods about the statistical tests that were applied to the data; this information is only included in figure legends and parenthetical remarks.

Response: Thank you for bringing up this point. We have made supplements in Materials and methods of the revised manuscript.

Line 459-461: The statistical analyses were performed using `wilcox.test` function of `vegan` package (45) in R (v4.0.4). The least-squares linear regression analysis was performed using `lm` function of `ggplot` package in R (v4.0.4).

Minor comments:

- The use of "clean" throughout to refer to quality filtered reads is confusing, since the deduplicated reads are also "cleaned." I would suggest changing the terminology to "standard" or similar to emphasize that your key advance is the deduplication, rather than the cleaning process.

Response: Thanks for your suggestion. We have replaced "cleaned" with "standard" throughout the revised manuscript.

- Lines 176-178: This is confusing; I think what is meant here is that most of the MAGs recovered with the standard/"clean" assembly were also recovered after deduplication of reads, but that the new MAGs generated by the deduplication process

were not just duplicate species, they were new, not previously recovered species.

Response: Our apologies for making you confused. Your understanding is precisely right. We have revised the sentence to avoid further confusion.

Line 186-188: Most of the MAGs recovered from the standard data were also recovered from data with deduplication. In contrast, extra species-level MAGs were recovered from deduplicated data that were not recovered from standard data (Fig. S4).

- Lines 206-207: Consider revising language. Fewer chimeras being present implies that there were some MAGs that fully lacked disagreement, whereas here what is being assessed is a rate of disagreement on a gradient from fully distinct to "fully chimeric (not interpretable as a distinct organism/species-level genome)

Response: Thank you for bringing up this point. Our apologies for making you confused here. We intended to compare the proportions of MAGs that contains chimera between data with and without deduplication. We assessed the chimera presence in MAGs by evaluating the homogeneity of their taxonomic annotation for each MAG, as follows: "No annotation" if less than two marker genes were annotated; "Agreeing" if all marker genes had the same taxonomic annotation at all classification levels; and "Disagreeing" if different taxonomic annotation of marker genes presented at any classification level. The disagreement rate was calculated as the number of MAGs containing chimera divided by the total number of MAGs excluding "No annotation". We compared the disagreement rate at the different taxonomic levels. For a specific MAG, we first checked the taxonomic annotation of 10 marker genes at the phylum level. If more than two marker genes were found in the MAGs and they have the same taxonomic annotation at the phylum level, the MAG will be considered as "agreeing"; otherwise, the MAG will be regarded as "disagreeing". Then we will check the disagreement rate at other taxonomic levels, such as class, order, family and genus. We have revised the language in the Results of the manuscript.

Line 229-237: The reliance on the counts of single-copy marker genes for

contamination estimation, as implemented in the ChekM (19), can still lead to chimerism, with populations sometimes from entirely different clades being mixed within a single genome (20). To check the chimera risk for the MAGs constructed in this study, we annotated 10 single-copy marker genes that were validated by mOTUs profiler (21) for their accuracy in genome taxonomic classification and assessed the presence of chimera in MAGs by evaluating the homogeneity of their taxonomic annotation for each MAG. We then compared the proportions of MAGs (disagreement rates) that contains chimera between data groups with and without deduplication.

- Lines 255-256: Consider expanding this list of references; this would be more impactful if a more representative set of studies was included.

Response: Thanks for your suggestion. We have expanded the list of references, including glacier, activated sludge and sediment microbiomes as shown in Line 282 of the revised manuscript.

- Lines 284-286: It is unclear how the duplicated reads and skewed coverage information specifically leads to contamination in this explanation. Why would it not affect only yields, beyond your observations?

Response: Thank you for bringing up this point. The reasonable explanation is that the binning tools may be misled by the skewed coverage information of contigs and therefore cluster the contigs into the MAGs, which they shouldn't belong to. We have also added an explanation in Line 288-289 of the revised manuscript.

Line 208-310: Therefore, duplicate reads might distort coverage abundance profiles of contigs, which might mislead the binning tools into clustering the contigs into MAGs that they should not belong to, leading to the increased contamination level of MAGs.

- Lines 293-294: Is there a consensus between binning developers, or could this be specifically associated with the binning tool you used in the study?

Response: Thank you for bringing up this point. The cross-sample binning is recommended by the developers of all the mainstream binning tools, such as

MetaBAT2, CONCOCT and Maxbin2. For example, the authors of MetaBAT2 ⁶ stated in their paper, “where $w = n_{ABD}/(n_{ABD} + 1)$, and n_{ABD} represents the number of effective samples which have enough coverage (by default >1) for at least one of the contigs. When the number of samples increases, ABD becomes more reliable, so w effectively decreases the relative weight of TNF and increases the weight of ABD. Whenever there are three or more samples available, an ABD correlation score (COR) is also calculated using the Pearson correlation coefficient and then rank-normalized using ABD.”. The authors of CONCOCT ⁷ stated in their paper, “Here we present CONCOCT, a program that uses Gaussian mixture models to cluster contigs into genomes based on sequence composition and coverage across multiple samples.”.

- Lines 326-328: Can duplicate reads be wholly avoided? Is there a recommendation for assessment for identifying when duplicate reads may exist?

Response: Regretfully, up to now, the artificial duplicate reads could not be wholly avoided in metagenomic sequencing. The artificial duplicate reads may consist of PCR duplicates arising from library preparation and optical duplicates, clustering duplicates and sister duplicates arising from paired-end sequencing. It may be reduced to an extremely level with sequencing technology advanced.

FastQC could be used to evaluate the duplicates rate of metagenomic samples.

- Lines 343-344: parameters used to decide on which reads were eliminated, as well as a list of adapters, should be provided (either in the text or in a public repository).

Response: Thank you for bringing up this point. The Fastp was used for deduplication in this study. The adapters can be detected by per-read overlap analysis, which seeks for the overlap of each pair of reads. This method is robust and fast, so we usually don't need to input the adapter sequences. If the number of “N” accounts for more than 10% of the total read bases, the paired reads will be removed. If the low-quality ($Q \leq 5$) bases account for more than 50% of the total read bases, the paired reads will be removed. The parameters we used for quality control and deduplication in this

study are as follows: “--qualified_quality_phred 5”, “--unqualified_percent_limit 50”, “--n_base_limit 15”, “--dedup”, and “--dup_calc_accuracy 3”. The accuracy level to calculate duplication ranges from 1 to 6, and “3” is the default value for deduplication. The higher level uses more memory and more time. We also provided these parameters in the Line 395-410 of the revised manuscript.

- Lines 350-353: for the cross-sample coverage, it is unclear what method was used to calculate coverage profiles. If this was using the companion script to MetaBAT2, this should be specified and the workflow used to generate bam coverage files should also be included.

Response: Thank you for bringing up this point. We used the Metawrap pipeline for the binning of assemblies, which was clarified in the revised manuscript (Line 408). The script “jgi_summarize_bam_contig_depths” within MetaBAT2 was used for coverage calculation.

- Line 354: parameters/settings used for MetaBAT2?

Response: We used the default parameters of MetaBAT2 for binning. We have clarified this in the revised manuscript in the Line 415-416.

Line 415-416: The coverage profiles of each assembly were then used to inform automated binning of MetaBAT2 (v2.12.1) (49) with the default parameters “--minContig 1500 --maxP 95 --minS 60 --maxEdges 200 --minClsSize 200000 --minCV 1 --minCVSum 1”.

- Lines 364-365: Do you expect all MAGs to be prokaryotic? Details on the filtering protocol used are missing from the sequencing methodology.

Response: Thank you for bringing up this point. The MAGs are all prokaryotic genomes, as the CheckM we used to evaluate genomes’ completeness and contamination is designed for prokaryotic genomes, i.e., bacteria and archaea⁸. The MAGs were filtered with a quality over 50%, in which the quality was calculated as completeness–5×contamination. We also clarified this point in the Line 433 in the

revised manuscript.

- Lines 376-377: can the in-house script be published alongside the paper?

Response: Thank you for bringing up this point. We used “time” function (`\time -f %e -v -o Running.log`) in Linux shell to record memory requirement and elapsed time. Then, each sample's log file was merged using the “paste” function in Linux shell. We also made supplements in the revised manuscript.

Line 455-457: Elapsed time and RAM taken by the software to complete the metagenomic assembly and binning were recorded using “time” function in Linux shell with customized options: `\time -f %e -v -o Running.log`.

- Lines 370-372: How were these 10 marker genes selected? Can we be sure that they are balanced?

Response: Thank you for bringing up this point. The 10 marker genes were selected from the mOTUs profiler⁹⁻¹¹, a widely-used tool for microbial community profiling. The 10 marker genes chosen have the properties of universal occurrence and low rate of horizontal gene transfer, meaning that they have good phylogenetic properties and can be used to track the evolutionary relationships between different microorganisms accurately. We also made supplements to the Materials and methods of the revised manuscript.

Line 444-448: We annotated 10 single-copy marker genes selected from mOTUs profiler (20). The 10 marker genes chosen have the properties of universal occurrence and a low rate of horizontal gene transfer, meaning that they have good phylogenetic properties and can be used to accurately track the evolutionary relationships between different microorganisms (20,55).

- Line 378+: Details on how figures were generated is missing; were all statistical tests and figures generated using base R utilities? If so, it would be good to explicitly specify this, as well as to list functions used in R for statistical analyses.

Response: Thanks for your kindly reminding. Yes, all statistical tests and figures in

this study are generated using package in R. We have made supplements in the revised manuscript.

Line 457-461: The figure of estimated average coverage was generated using Nonpareil package (18) in R (v4.0.4), and other figures were plotted using the ggplot package in R (v4.0.4). The statistical analyses were performed using wilcox.test function of vegan package (45) in R (v4.0.4). The least-squares linear regression analysis was performed using lm function of the ggplot package in R (v4.0.4).

- Figure 2: It appears as though the significance test is being calculated across samples, yet bars are used to show the outcomes of individual assemblies for each environmental sample. This figure would be much easier to read and interpret if boxplots or other distribution-based plots could compare each set of environmental samples, rather than distinct bars that do not appear to be ordered in any particular way from left to right (unless this information is missing from the figure).

Response: Thank you for bringing up this point and for your suggestion. We have replotted the figure, as you suggested, which is much easier to read and interpret.

- Figure 3: A scatter plot with colors or shapes indicating each environmental sample type (along with a one-to-one line) may make it easier to read and interpret the difference in number of MAGs before and after deduplication. The two adjacent bars are similar in size and it is challenging to compare the relative difference between each sample type

Response: Thanks for your suggestion. We have replotted the Figure 3 as you suggested. We used the boxplot with scattered points to show the figure with colors indicating each sample type. We also added the one-to-one lines to indicate the change between paired samples (with deduplication and without deduplication)

Reference

1. Niu, B., et al., Artificial and natural duplicates in pyrosequencing reads of metagenomic data. *BMC Bioinformatics* **2010**, *11*, (1), 187.
2. Smith, E. N., et al., Biased estimates of clonal evolution and subclonal heterogeneity can arise from PCR duplicates in deep sequencing experiments. *Genome Biology* **2014**, *15*, (7), 420.
3. Chafee, M., et al., The effects of variable sample biomass on comparative metagenomics. *Environmental Microbiology* **2015**, *17*, (7), 2239-2253.
4. Parkinson, N. J., et al., Preparation of high-quality next-generation sequencing libraries from picogram quantities of target DNA. *Genome Research* **2012**, *22*, (1), 125-133.
5. Paoli, L., et al., Biosynthetic potential of the global ocean microbiome. *Nature* **2022**, *607*, (7917), 111-118.
6. Kang, D. D., et al., MetaBAT 2: an adaptive binning algorithm for robust and efficient genome reconstruction from metagenome assemblies. *PeerJ* **2019**, *7*, e7359.
7. Alneberg, J., et al., Binning metagenomic contigs by coverage and composition. *Nature Methods* **2014**, *11*, (11), 1144-1146.
8. Parks, D. H., et al., CheckM: assessing the quality of microbial genomes recovered from isolates, single cells, and metagenomes. *Genome Research* **2015**, *25*, (7), 1043-1055.
9. Milanese, A., et al., Microbial abundance, activity and population genomic profiling with mOTUs2. *Nature Communications* **2019**, *10*, (1), 1014.
10. Sunagawa, S., et al., Metagenomic species profiling using universal phylogenetic marker genes. *Nature Methods* **2013**, *10*, (12), 1196-1199.
11. Ruscheweyh, H.-J., et al., Cultivation-independent genomes greatly expand

taxonomic-profiling capabilities of mOTUs across various environments.

Microbiome **2022**, *10*, (1), 212.

January 15, 2023

Prof. Feng Ju
Westlake University
Hangzhou
China

Re: Spectrum04282-22R1 (Deduplication Improves Cost-Efficiency and Yields of De novo Assembly and Binning of Shot-Gun Metagenomes in Microbiome Research)

Dear Prof. Feng Ju:

Thank you for submitting your manuscript to Microbiology Spectrum. Your manuscript has been read again by the two reviewers, and both of them are satisfied with your revision. You could also find some additional but minor comments below this message. I expect that your manuscript is very close to acceptance. When submitting the revised version of your paper, please provide (1) point-by-point responses to the issues raised by the reviewers as file type "Response to Reviewers," not in your cover letter, and (2) a PDF file that indicates the changes from the original submission (by highlighting or underlining the changes) as file type "Marked Up Manuscript - For Review Only". Please use this link to submit your revised manuscript - we strongly recommend that you submit your paper within the next 60 days or reach out to me. Detailed instructions on submitting your revised paper are below.

Link Not Available

Sincerely,

Jianjun Wang

Journals Department
Reviewer comments:

Reviewer #1 (Comments for the Author):

The authors have addressed all my concerns

Reviewer #2 (Comments for the Author):

Thank you for addressing the comments I and the other reviewer made on your manuscript. I am satisfied with the amendments that you made, and find the changes that you made to the discussion on the taxonomy of the identified MAGs to be fascinating. I still do think that it is appropriate to explicitly reference the use of companion scripts, e.g. `jgi_summarize_bam_contig_depths`.

However, I very much appreciate the addition of many technical and parameter details, and find the figure changes to be very helpful.

Staff Comments:

Preparing Revision Guidelines

Please return the manuscript within 60 days; if you cannot complete the modification within this time period, please contact me. If you do not wish to modify the manuscript and prefer to submit it to another journal, please notify me of your decision immediately so that the manuscript may be formally withdrawn from consideration by Microbiology Spectrum.

Responses to the Comments of the Editor and Reviewers

Dear Editor and Reviewers,

Thanks for your evaluation on our paper entitled “Deduplication Improves Cost-Efficiency and Yields of *De novo* Assembly and Binning of Shot-Gun Metagenomes in Microbiome Research” (Manuscript ID: Spectrum04282-22R1). We have revised the manuscript according to your valuable comments. Please see the revision highlighted in yellow in the revised manuscript and below the point-by-point responses to the comments.

Editor’s comments:

Thank you for submitting your manuscript to *Microbiology Spectrum*. Your manuscript has been read again by the two reviewers, and both of them are satisfied with your revision. You could also find some additional but minor comments below this message. I expect that your manuscript is very close to acceptance. When submitting the revised version of your paper, please provide (1) point-by-point responses to the issues raised by the reviewers as file type "Response to Reviewers," not in your cover letter, and (2) a PDF file that indicates the changes from the original submission (by highlighting or underlining the changes) as file type "Marked Up Manuscript - For Review Only". Please use this link to submit your revised manuscript - we strongly recommend that you submit your paper within the next 60 days or reach out to me. Detailed instructions on submitting your revised paper are below.

Response: Thank you for your efforts on processing the manuscript. We have revised the manuscript according to the reviewers’ comments. We also prepared and submitted the materials of the revised paper as you suggested.

Reviewer #1:

The authors have addressed all my concerns

Response: Thank you very much for your professional evaluation and recognition of our work.

Reviewer #2:

Thank you for addressing the comments I and the other reviewer made on your manuscript. I am satisfied with the amendments that you made, and find the changes that you made to the discussion on the taxonomy of the identified MAGs to be fascinating. I still do think that it is appropriate to explicitly reference the use of companion scripts, e.g., `jgi_summarize_bam_contig_depths`. However, I very much appreciate the addition of many technical and parameter details, and find the figure changes to be very helpful.

Response: Thanks very much for your efforts on the evaluation of our work. We are appreciated that the revised manuscript addressed your concerns. Following your suggestion, we have added explicitly reference of the script used for coverage calculation as shown in Line 412-413 of the revised manuscript.

Line 412-413: using the script “`jgi_summarize_bam_contig_depths`” (<https://bitbucket.org/berkeleylab/metabat>).

January 18, 2023

Prof. Feng Ju
Westlake University
Hangzhou
China

Re: Spectrum04282-22R2 (Deduplication Improves Cost-Efficiency and Yields of De novo Assembly and Binning of Shot-Gun Metagenomes in Microbiome Research)

Dear Prof. Feng Ju:

Your manuscript has been accepted, and I am forwarding it to the ASM Journals Department for publication. You will be notified when your proofs are ready to be viewed.

Sincerely,

Jianjun Wang
Editor, Microbiology Spectrum
